# GRAPH-ASSISTED PREDICTIVE STATE REPRESENTATIONS FOR MULTI-AGENT PARTIALLY OBSERVABLE SYSTEMS

**Zhi Zhang**[1]    **Zhuoran Yang**[2]    **Han Liu**[3]    **Pratap Tokekar**[4]    **Furong Huang**[5]
[1,4,5]University of Maryland, College Park    [2]Yale University    [3]Northwestern University
[1]`zhizhang5763@gmail.com`, {[4]`furongh`,[5]`tokekar`}`@umd.edu`
[2]`zhuoran.yang@yale.edu`, [3]`hanliu@northwestern.edu`

## ABSTRACT

We study reinforcement learning for partially observable multi-agent systems where each agent only has access to its own observation and reward and aims to maximize its cumulative rewards. To handle partial observations, we propose graph-assisted predictive state representations (GAPSR), a scalable multi-agent representation learning framework that leverages the agent connectivity graphs to aggregate local representations computed by each agent. In addition, our representations are readily able to incorporate dynamic interaction graphs and kernel space embeddings of the predictive states, and thus have strong flexibility and representation power. Based on GAPSR, we propose an end-to-end MARL algorithm that simultaneously infers the predictive representations and uses the representations as the input of a policy optimization algorithm. Empirically, we demonstrate the efficacy of the proposed algorithm provided on both a MAMuJoCo robotic learning experiment and a multi-agent particle learning environment.

## 1 INTRODUCTION

In partially observable decision-making systems, it is pivotal for the agents to infer the latent state of the system from past observations. The predictive state representation (PSR) (Littman et al., 2001) is a representation of the history by a vector of predictions of set of tests conditioning on the history. In particular, a history is a sequence of part observations and actions. A test is sequences of future actions and future observations, which is true if and only if all the observations occur, given that all the actions taken. Each entry of the predictive state of representation of is given by the conditional probability that the corresponding test holds true given that history. A fundamental assumption of PSR model is that these representations of the history are a sufficient statistics for predicting future observations and thus are able to represent the underlying state which is unobserved (Littman et al., 2001; Hefny et al., 2015; Sun et al., 2016). A key feature of PSR is that it only involves observable actions and observations and also encodes the latent state. Thus, the system dynamics can be recovered from data without resorting to the estimation of the unobserved latent states. Furthermore, using kernel embeddings of conditional distributions, PSR can be embedded into reproducing kernel Hilbert spaces (RKHS) (Boots et al., 2013), which further enhances the representation powerful of PSR. Furthermore, Littman et al. (2001); James & Singh (2004); Singh et al. (2012) shows that PSR offers an alternatively more compact state representation than POMDP models. Therefore, PSR is considered as a powerful representation learning tool and has been used as an indispensable component of various model-based single-agent reinforcement learning methods (James & Singh, 2004; Boots et al., 2011; 2013; Hamilton et al., 2014; Sun et al., 2016; Hefny et al., 2015; 2018a).

Furthermore, when it comes to multi-agent systems with partial observations, is desirable to construct predictive states for multi-agent reinforcement learning (MARL). However, directly extending single-agent PSR would require building tests based on joint observations and joint actions. As a result, learning the PSR involves tensor decomposition which is computationally prohibitive (Chen et al., 2020). Moreover, another drawback of such an approach is that oftentimes the agents only have local observations. Furthermore, in multi-agent systems, the observations of each agent are influenced by the actions of other agents. As a result, the representations should also take the other agents into account. Specifically, we aim to address the following question:

Can we construct multi-agent predictive representations based on local information for reinforcement learning?

In this work, we provide a positive answer to this question. Specifically, we propose a new framework of multi-agent PSR named as Graph-Assisted Predictive State Representations (**GAPSR**), which utilizes local information possessed by each agent to construct scalable representations. Specifically, we assume each agent has local observations but is able to observe joint actions. Let $n$ be the number of agents. For any agent $i$, we first construct $n$ primitive interaction predictions $\{\mathbf{q}_{i,j}\}_{j \in [n]}$ via single-agent PSR, where $\mathbf{q}_{i,j}$ is constructed using agent $i$'s local observations and agent $j$'s local actions. Then agent $i$'s final predictive representations $\mathbf{q}_i$ is obtained by aggregating $\{\mathbf{q}_{i,j}\}_{j \in [n]}$ according to the underlying agent connectivity graph which characterize the proximity among agents. Finally, these graph assisted representations are used as the proxy of state variable and fed into a standard MARL algorithm, e.g., multi-agent actor-critic for decision making.

Our GAPSR framework enjoys a few desired properties. First, GAPSR aggregates the information about other agents via the agent connectivity graph, which encodes the interaction among agents. Such an aggregation mechanism enables us to implement GAPSR on a single-agent level while maintaining other agents' interactions through the encoding of the interactive graph topology. Second, the GAPSR of each agent involve the actions of all the other agents connected to that agent. In other words, GAPSR correctly captures the non-stationarity caused by the other agents' actions. Besides, the agent connectivity graphs can take various forms. In particular, we build our model under three common graphs, namely static complete graphs, static non-complete graphs, and dynamic graphs, covering as many real-world scenarios as possible. Third, utilizing kernel embedding of conditional distributions, the predictive representations lie in RKHS with strong representation power. Finally, GAPSR are readily to be incorporated into any state-of-the-art MARL algorithm as a proxy of the state variable in an end-to-end fashion. Thus, we believe that GAPSR could be a promising representation learning framework for multi-agent partially observable systems.

Finally, we implement an instantiation of the proposed algorithm that combines GAPSR with multi-agent actor-critic, where we simultaneously learn GAPSR and the agents' policies in an end-to-end manner with fully differentiable neural network structures. We test our algorithm through systematic numerical experiments on MAMuJoCo robotic learning experiments (de Witt et al., 2020) and multi-agent particle learning environments (Ryu et al., 2020), and compare our proposed method against two baselines as detailed in Section 6. The results demonstrate the efficiency of our method.

## 2 RELATED WORKS

**Partially Observable Environment.** Real-world agents often experience situations that the observed signals are aliased and do not fully determine their state in the system. This is particularly true for multiple agents environments where agents have partial observability due to limited communication (Oliehoek & Amato, 2016). In accommodation to the partially observable environment, POMDP has been adopted by Kaelbling et al. (1998), and the algorithms (Kaelbling et al., 1998; Cassandra, 1998; Thrun, 1999; Pineau et al., 2003; Poupart & Vlassis, 2008; Platt Jr et al., 2010) for determining an optimal policy have shifted to using the probability distributions (belief state) over the state space instead of exact state space. In general, they have high complexity or suffer from local optima. Moreover, the most common POMDP policy learning assumes the agent has access to a priori knowledge of the system. The access to such prior knowledge has a premise that the agent has considerable domain knowledge (Kaelbling et al., 1998). However, it is expected that the real-world agents learn the system model and thus a planning policy further without knowledge of the domain.

**Overview of PSR.** Littman et al. (2001); James & Singh (2004) introduced the PSR over an expressive and robust framework for modeling dynamical systems and defined PSR as a representation of state by using a vector of predictions of fully observable quantities (tests) conditioned on past events (histories). A predictive model is constructed directly from execution traces in the PSR framework, utilizing minimal prior information about the domain. The PSR paradigm subsumes POMDP as a special case (Littman et al., 2001). PSR is considered much more compact than POMDP (Aberdeen et al., 2007). The spectral learning method has been proved to show success for learning the PSR (Boots et al., 2011; Jiang et al., 2018). There are other classes of dynamical system learning algorithms that are based on likelihood-based optimization or sampling approaches (Frigola et al., 2013), but they are prone to poor local optima. The spectral learning represents the estimated state by sufficient statistics of future observations and estimates the model parameters by method of moments. However,

this line of algorithms is hard to incorporate prior information (Hefny et al., 2015). Thus, Hefny et al. (2015; 2018a) introduce the supervised learning method to learn PSR and proves its convergence. Although many works study PSR in discrete action space (Hsu et al., 2012; Siddiqi et al., 2010; Boots et al., 2011), Boots et al. (2013) propose Hilbert space embedding (HSE)-PSR to deal with continuous actions. Hefny et al. (2018a) use an approximation of HSE-PSR by Random Fourier transform (RFF) and built a more principled generalization of PSR to deal with high dimensions. However, all of these studies aim for the single agent scenario.

**PSR and RL.** The predictive states estimated by the PSR are considered as states in a fully observable Markov Decision Process so that the value function is learned on these states. This line of work has been done in the single-agent environment (Boots & Gordon, 2010; Boots et al., 2011; Hamilton et al., 2014; Venkatraman et al., 2017; Hefny et al., 2018b). Especially, Hefny et al. (2018b) propose the recurrent predictive state in the RNN network. Moreover, the learning PSR and policy functions are connected with the end-to-end training.

**MAPSR and MARL.** A close related work Chen et al. (2020), which proposes a multi-agent PSR model that is formulated as a $n$-way tensor, where $n$ is the number of agents. As a result, learning the parameters is achieved by decomposing the $n$-way tensor, which becomes computationally prohibitive when the number of agents is large. Moreover, it seems challenging to incorporate benign structures such as sparsity into the model, as doing so leads to a more complicated set of moment equations. Furthermore, our work belongs to the vast literature on MARL for (partially observable) Markov Games. See, e.g., Lyu & Amato (2020); Son et al. (2019); Zhang et al. (2019); Rashid et al. (2018); Foerster et al. (2018); Lowe et al. (2017); Baker et al. (2019); Yu et al. (2021) and the references therein. Also see Zhang et al. (2021); Gronauer & Diepold (2022); Canese et al. (2021) for recent surveys on MARL. Recent works propose more sophisticated deep MARL algorithms for multi-agent problems under the paradigm of centralized training with decentralized execution (Zhou et al., 2020; Sunehag et al., 2017; Lowe et al., 2017; Foerster et al., 2018; Rashid et al., 2018). Our work seems not directly comparable to these works as our goal is to learn good representations in the partially observable multi-agent systems via model-based RL, whereas most of these works adopt model-free methods or focus on Markov games without partial observations.

## 3 BACKGROUND: SINGLE-AGENT PSR

In this section, we introduce the background on single-agent PSR. Also see §A.1 for a more detailed introduction.

**Predictive State Representation (PSR).** A prediction of a state is defined as the conditional probability of seeing a test's observations in sequence given that actions of the test are taken in sequence from a history (Littman et al., 2001). Givn a finite observation space $\mathcal{O}$ ($o \in \mathcal{O}$) and action space $\mathcal{A}$ ($a \in \mathcal{A}$). A **test** of length $k$ at time $t$, is defined as a sequence of action-observation pairs that starts at time $t$ and ends at time $t+k-1$, $\{(o_l, a_l)\}_{l=t}^{t+k-1} = \{o_t, a_t, o_{t+1}, a_{t+1}, \cdots, o_l, a_l, \cdots, o_{t+k-1}, a_{t+k-1}\}$. A **history**, at time $t$ is a sequence of action-observation pairs that start from the beginning of time and ends at time $t-1$, $\{(o_l, a_l)\}_{l=1}^{t-1} = \{o_1, a_1, \cdots, o_l, a_l, \cdots, o_{t-1}, a_{t-1}\}$.

**Hilbert Space Embeddings of Predictive State Representations.** In this work, we are interested in extending PSR to decision makings of controlled systems with continuous actions. So we use the model introduced by Hefny et al. (2018a). In this model, predictive state $\mathbf{q}_t$ and extended predictive state $\mathbf{p}_t$ satisfies

$$\mathbf{q}_t \psi_t^a = \mathbb{E}[\psi_t^o | \psi_t^a; \psi_t^h], \quad \mathbf{p}_t \xi_t^a = \mathbb{E}[\xi_t^o | \xi_t^a; \psi_t^h], \tag{1}$$

(i.e., $\mathbf{q}_t$ and $\mathbf{p}_t$ are conditional linear expectation operators which maps to the conditional expectation of future observations), where

$$\psi_t^o := \phi_O(\{o_l\}_{l=t}^{t+k-1}), \quad \psi_t^a := \phi_A(\{a_l\}_{l=t}^{t+k-1}), \tag{2}$$

are feature maps by kernels $k_O$, $k_A$ over future observation and action features. The extended predictive state compared to predictive state adds one more pair of $\{(a_{t+k}, o_{t+k})\}$ to the prediction. The $\xi_t^a$ and $\xi_t^o$ are the corresponding extended feature maps, which satisfy

$$\xi_t^o = \psi_t^o \otimes \phi_t^o, \quad \xi_t^a = \psi_t^a \otimes \phi_t^a. \tag{3}$$

Here $\phi_t^o := \phi_o(o_t)$ and $\phi_t^a := \phi_a(a_t)$ are the shorthands for one time feature map by $k_o$ and $k_a$. We use $\otimes$ to denote the transposed Khatri–Rao product for two matrices with the same number of rows, and each row of the resultant matrix is the vectorzied outer product of the corresponding row vectors

in the two matrices. Also, we use

$$\psi_t^h := \psi^h(\{(o_l, a_l)\}_{l=1}^{t-1}) \tag{4}$$

to define a set of features extracted from previous observations and actions (typically from a fixed length window ending at $t - 1$).

## 3.1 Notation of PSR For Multi-Agent setting

In the subsequent section, we will bring the HSE-PSR model to the multi-agent settings. In the following, we introduce the notation for the multi-agent setup. For any agent $i$ in an $n$-agent system, given a set of kernels $\{k_{O_i}\}_{i=1}^n, \{k_{A_i}\}_{i=1}^n, \{k_{o_i}\}_{i=1}^n, \{k_{a_i}\}_{i=1}^n$ for every agent, we define agent $i$'s feature maps using the single-agent PSR presented in the previous section. In particular, for all $i \in [n]$, we let $\psi_{t,i}^o$ and $\psi_{t,i}^a$ be defined as in (2) using agent $i$'s local observations and actions, respectively. Similarly, we let $\xi_{t,i}^o$ and $\xi_{t,i}^a$ be defined in (3), and let $\phi_{t,i}^o$ and $\phi_{t,i}^a$ be defined using $\phi_{o_i}$ and $\phi_{a_i}$, respectively. Furthermore, for all $i, j \in [n]$, we let $\psi_{t,i,j}^h$ be defined as in (4) using the observations of agent $i$ and the actions of agent $j$. Moreover, we use the $\mathbf{q}_{t,i}$ and $\mathbf{p}_{t,i}$ to denote its predictive state and extended predictive state, whose definitions will be presented in the next section.

## 4 Dynamic Interaction Graph For GAPSR Model

Many works have considered graph representation of the multi-agent network (Liu et al., 2020b; Ryu et al., 2020). In general, the relationship between agents is characterized by an undirected graph. We introduce a dynamic interaction graph to represent the GAPSR by considering the interaction between agents.

**Definition 1.** *Let $G = (V, E)$, including the set $V$ of nodes and set $E$ of the edges. Each node represents the agent entry, and the edge represents the relationship between the two adjacent agents.*

Here we suppose the graph structure is given, that means the number of nodes ($n$), the number of edges ($m$), the edge weights, and the maximum number of degrees ($k$) are available to us. We think that this kind of presupposition is very reasonable because in the real world, for example, there are multiple robots; we can quickly get the geographic position of the robots through sensors, then calculate the structure of the graph formed by them.

## 4.1 Static Complete Graph

We starts with a static complete graph $G_c$, where the relationship between nodes are invariant to time change. A complete graph has $m = n(n-1)/2$ edges, where $m = |E|$. For each agent, we can represent its PSR by considering other agents' interactions by

$$\mathbf{q}_{t,i} := g\left(\{\mathbf{q}_{t,i,j}\}_{j=1}^n\right) = \sum_j \mathbf{q}_{t,i,j}, \tag{5}$$

same for $\mathbf{p}_{t,i}$. To consider the interactive behavior between agents, we introduce two additional notations $\mathbf{q}_{t,i,j}$ and $\mathbf{p}_{t,i,j}$. Let $\{(i,j)\}_{i,j=1}^n$ represents a pair of agents on a $n$ multi-agent system. The $\mathbf{q}_{t,i,j}$ is a primitive predictive state of $i$'s observation ($\psi_{t,i}^o$) by intervening agent $j$'s action ($\psi_{t,j}^a$) and observing agent $i$'s observation history and agent $j$'s action history ($\psi_{t,i,j}^h$), and $\mathbf{p}_{t,i,j}$ is the extended counterpart, where they satisfy the relationship for single agent case in (1)

$$\mathbf{q}_{t,i,j}\psi_{t,j}^a = \mathbb{E}[\psi_{t,i}^o | \psi_{t,j}^a; \psi_{t,i,j}^h], \quad \mathbf{p}_{t,i,j}\xi_{t,j}^a = \mathbb{E}[\xi_{t,i}^o | \xi_{t,j}^a; \psi_{t,i,j}^h]. \tag{6}$$

If $i = j$, then it becomes an exact single agent scenario. Similarly, we use the same approach to represent $\mathbf{p}_{t,i}$ as shown in (5). Each agent's PSR $\mathbf{q}_{t,i}$ and extended PSR $\mathbf{p}_{t,i}$ are modeled by fully considering all other available agents.

Based on equation (5), in practice, estimating $\mathbf{q}_{t,i}$ and $\mathbf{p}_{t,i}$ denoted as $\widehat{\mathbf{q}}_{t,i}$ and $\widehat{\mathbf{p}}_{t,i}$ requires us to get $\widehat{\mathbf{q}}_{t,i,j}$ and $\widehat{\mathbf{p}}_{t,i,j}$ at first.

### 4.1.1 Estimation of $\mathbf{q}_{t,i,j}$ and $\mathbf{p}_{t,i,j}$

To estimate the $\mathbf{p}_t$ and $\mathbf{q}_t$ for just one agent, Hefny et al. (2018a) use the supervised learning method. They show that $\widehat{\mathbf{q}}_t = M_{\psi_t^o | \psi_t^a; \psi_t^h} = C_{\psi_t^o \psi_t^a | \psi_t^h}(C_{\psi_t^a \psi_t^a | \phi_t^h} + \lambda I)^{-1}$, where $M_{A|B;c}$ is a linear operator that satisfies $\mathbb{E}[A|B = b; C = c] = M_{A|B;c}b$, and $C_{XY} := \mathbb{E}[\phi(X) \otimes \phi(Y)]$ is the uncentered covariance operator, and $C_{XY|z}$ is covariance of $X$ and $Y$ given $Z = z$. They estimate $C_{\psi_t^o \psi_t^a | \psi_t^h}$, $C_{\psi_t^a \psi_t^a | \phi_t^h}$ by sampling data, then get the estimation $\widehat{\mathbf{q}}_t$. The same procedure is used to estimate $\widehat{\mathbf{p}}_t$ by replacing the features $\psi_t^o$ and $\psi_t^a$ with their extend counterparts $\xi_t^o$ and $\xi_t^a$.

Similar to the single-agent case above, the representation of $\mathbf{q}_{t,i,j}$ can be achieved as

$$\widehat{\mathbf{q}}_{t,i,j} = M_{\psi_{t,i}^o|\psi_{t,j}^a;\psi_{t,i,j}^h} = C_{\psi_{t,i}^o\psi_{t,j}^a|\psi_{t,i,j}^h}\left(C_{\psi_{t,j}^a\psi_{t,j}^a|\phi_{t,i,j}^h} + \lambda I\right)^{-1}. \tag{7}$$

To estimate $C_{\psi_{t,i}^o\psi_{t,j}^a|\psi_{t,i,j}^h}$ and $C_{\psi_{t,j}^a\psi_{t,j}^a|\psi_{t,i,j}^h}$, we learn two linear maps $T_{i,j}$ and $U_{i,j}$ such that $C_{\psi_{t,i}^o\psi_{t,j}^a|\psi_{t,i,j}^h} \approx T_{i,j}\left(\psi_{t,i,j}^h\right)$ and $C_{\psi_{t,j}^a\psi_{t,j}^a|\psi_{t,i,j}^h} \approx U_{i,j}\left(\psi_{t,i,j}^h\right)$. The training examples for $T_{i,j}$ and $U_{i,j}$ consist of pairs $(\psi_{t,i,j}^h, \psi_{t,i}^o \otimes \psi_{t,j}^a)$ and $(\psi_{t,i,j}^h, \psi_{t,j}^a \otimes \psi_{t,j}^a)$. The learning of $\widehat{\mathbf{p}}_{t,i,j}$ can be done in a similar way. More details can be found in §A.2.1

After calculation of $\widehat{\mathbf{q}}_{t,i,j}$ and $\widehat{\mathbf{p}}_{t,i,j}$, under the static complete graph setting where the interaction is considered, defined in equation (5), we can get the estimate of $\widehat{\mathbf{q}}_{t,i}$ and $\widehat{\mathbf{p}}_{t,i}$.

### 4.1.2 Theoretical Guarantee of Estimation of $\mathbf{q}_i$ and $\mathbf{p}_i$ Under the Static Complete Graph

Theoretically, we show that the difference between $\mathbf{q}_i$ and its estimator $\widehat{\mathbf{q}}_i$ is bounded with high probability.

**Theorem 1.** *Let $\pi_\Theta$ be a data collection policy and $\mathcal{H}$ is the range of $\pi_\Theta$ on joint histories. If Equation (5) and (7) used, then for all $h \in \mathcal{H}$ and any $\epsilon \in (0,1)$, such that $N > \frac{t_{A_j}^2 \log(2d_{A_j}/\epsilon)}{v\left(C_{\psi_j^a}\right)}$ where $N$ is the number of time points we collect sample, then $\|\widehat{\mathbf{q}}_i - \mathbf{q}_i\|$ is bounded as below with probability at least $1 - 3\epsilon$,* $\qquad \|\widehat{\mathbf{q}}_i - \mathbf{q}_i\| \leq n\Delta,$ (8) *where*

$$\Delta = \sqrt{\frac{u\left(C_{\psi_i^o|\psi_{i,j}^h}\right)}{v\left(C_{\psi_j^a|\psi_{i,j}^h}\right)^3}} \cdot \frac{\|\Delta_1\|^2 + 2u\left(C_{\psi_j^a|\psi_{i,j}^h}\right)\|\Delta_1\| + \lambda}{v\left(C_{\psi_j^a|\psi_{i,j}^h}\right)(1-\gamma)+\lambda} + \frac{\left\|C_{\psi_i^o\psi_j^a|\psi_{i,j}^h}\right\|\|\Delta_1\|+\|\Delta_2\|\left\|C_{\psi_j^a|\psi_{i,j}^h}\right\|+\|\Delta_2\|\|\Delta_1\|}{v\left(C_{\psi_j^a|\psi_{i,j}^h}\right)^2(1-\gamma)^2+\lambda}.$$

*Here $\Delta_1$, $\Delta_2$ are two other relevant bounds, we provide them in §E.1. $u(\cdot), v(\cdot)$ denote the largest, smallest eigenvalue of a matrix. And $\gamma = \frac{t_{A_j}^2 \log(2d_{A_j}/\epsilon)}{v\left(C_{\psi_j^a}\right)N} < 1$ is a constant that depends on the magnitude of the norm of $\psi_j^a$ (we assume $\left\|\psi_j^a\right\| \leq t_{A_j}$), the dimension of $\psi_j^a$ ($d_{A_j}$), the (uncentered) covariances ($C_{\psi_j^a} := \mathbb{E}\left[\psi_j^a\psi_j^{aT}\right]$) and the sample size ($N$).*

Theorem 1 says we need at least $N$ samples for the bound in equation (8) to be valid. We give the proofs in §E. It is not hard to obtain a bound for $\mathbf{p}_i$ using the same approach. We omit that.

### 4.2 Extension to Static Non-Complete Graph

In large-scale multi-agent systems, the number of agents is large, and not all agents need to interact with each other. A static non-completed graph can perfectly represent such a situation. For example, in a given static non-complete graph $G_s$, we know its maximum number of degree $k$, and we use the binary $n \times n$ matrix with each entry as $I_{i,j}$ to indicate the interaction between two agents. Then the GAPSR for each agent will be

$$\mathbf{q}_{t,i} := g\left(\{\mathbf{q}_{t,i,j}\}_{j=1}^n\right) = \sum_j I_{i,j}\mathbf{q}_{t,i,j}. \tag{9}$$

**Lemma 1.** *Under the same environment depicted in Theorem 1 and given a $G_s$ with $k$ maximum number of the degree to represent agents, then the bound in Theorem 1 can be rewritten as $\|\widehat{\mathbf{q}}_i - \mathbf{q}_i\| \leq k\Delta$.*

The conclusion of Lemma 1 is evident since we replace the $n$ with the $k$ neighbors, the total error bound is also decreased approximately as $\frac{k}{n}$.

### 4.3 Dynamic Graph

Real-world multi-agents can also formulate a time-dependent dynamic graph $G_d$, rather than a static graph. A dynamic graph has its structure dynamically changing with time. In other words, the edges can be inserted or deleted across time. The dynamic graph brings more challenges to the representation as the interaction relationship among agents changes constantly.

Braha & Bar-Yam (2009), Ma et al. (2017) and Zhao et al. (2010) consider a dynamic graph as a set of ordered static graphs. For each time point, we are given a static graph such that $G_d = \{G_{d1}, G_{d2}, \cdots, G_{dt}\}$, and a time-dependent given binary matrix with $I_{t,i,j}$ indicating the interaction between two agents at each time. Then we have the agent-wise PSR as

$$\mathbf{q}_{t,i} := g\left(\{\mathbf{q}_{t,i,j}\}_{j=1}^{n}\right) = \sum_j I_{t,i,j}\mathbf{q}_{t,i,j}. \tag{10}$$

Compared to (9), the coefficient $I_{t,i,j}$ is time-dependent, which brings a challenge to our theoretical bound. We consider a dynamic graph experiences a trajectory path, assuming every node has a chance at least $p$ to interact by connection with another node at any time point. For example, for a node $i$, if we take the union set of the nodes interacted with $i$ over the path, then the union set could form a static complete graph; in other words, if $i$ connects $j$, then we can obtain a valid sample to estimate $\mathbf{q}_{i,j}$ as the complete static graph does, if not, then we skip to the next time point. For the complete static graph, we need the trajectory to run at least $N$ time points to collect enough data to estimate our conditional operator $\mathbf{q}_{i,j}$ accurately. Furthermore, The total number of time points needed by $i$ until the $N^{th}$ interaction with $j$ follows a negative binomial distribution $\mathcal{NB}(N, p)$. On average, we need $\frac{N}{p}$ time points before we see $i, j$ completely connecting $N$ times. Now we consider node $i$ could interact with every node in a set of nodes ($J : |J| = n - 1$) for $N$ number of time points. We assume the chances being interacted between two nodes ($i, j \in J$) does not affect their interaction with other nodes. Thus we have a set of independent negative binomial random variables $\{J_l\}_{l=1}^{n-1} \sim \mathcal{NB}(N, p)$ to characterize the interaction of $i$ with $j \in J$. So we are interested in the expectation of the maximum of $J_1, \ldots, J_{n-1}$, a statistics that tells us the expected maximum number of time points of collection of measurements needed for the node $i$ to be able to connect with every node $j \in J$ for at least $N$ number of time points. We denote it as $J_{\{1,\ldots,n-1\}}$ and we have

$$\mathbb{E}\{J_{\{1,\ldots,n-1\}}\} = \mathbb{E}\{\max(J_1, \cdots, J_{n-1})\} = \sum_{N \geq 0} \left(q^N + Npq^{N-1} + \cdots + \binom{N}{N-1}p^{N-1}q\right)^{n-1}. \tag{11}$$

**Lemma 2.** *Under the same environment depicted in 1 and given a dynamic graph $G_d$ with every node has a chance at least $p$ to interact with another node in a one-time point, let $N$ be the number of time points we collect data of measurements in order to get the bound in Theorem 1, if Equation 10 and 7 used, then we need at least*

$$N' = \left(N - \frac{1}{2}\right) + K(q, n, N) - \frac{\gamma}{\log_{1/q}(1/q)} + F[K(q, n, N)] + \mathcal{O}(1), \tag{12}$$

*total number of time points, where $q = 1 - p$, $K(q, n, N) := \log_{1/q}(n - 1) + (N - 1)\log_{1/q}\left[\log_{1/q}(n - 1)\right] + (N - 1)\log_{1/q}p - \log_{1/q}(N - 1)!$, $F$ is a periodic $C^\infty$-function of period 1 and mean value 0 whose Fourier-coefficients are given by $\hat{F}(k) = -\frac{1}{\log(\frac{1}{q})}\Gamma(-\frac{2k\pi i}{\log(\frac{1}{q})})$ for $k \in \mathbb{Z} \setminus \{0\}$. Then $\|\hat{\mathbf{q}}_i - \mathbf{q}_i\|$ achieves the same bound as in Theorem 1 with probability at least $1 - 3\epsilon$. In other words, $\|\hat{\mathbf{q}}_i - \mathbf{q}_i\| \leq n\Delta$. Moreover, (12) is an asymptotic expansion of the right-hand of (11). We give the proof in §E.3.*

Lemma 2 gives the worst-case bound for our estimation under the dynamic graph. The result tells us that if we need 1 more sample of measurement for our algorithm to converge on the complete static graph, we need roughly $\log_{1/q}[\log_{1/q}(n - 1)]$ more samples on the dynamic graph. So as long as we allow enough learning time, the algorithm can converge with high probability.

**Complexity with Increasing Agents**. For the complete static graph, we need to evaluate $\hat{\mathbf{q}}_{i,j}$ and $\hat{\mathbf{p}}_{i,j}$ every time point, which requires $\mathcal{O}(n^2)$ operations and space. Overall, with an increased number of agents, our GAPSR has a polynomial $\mathcal{O}(n^2)$ scaled complexity, which is feasible for learning in a large number of agents environment and is more efficient compared to the centralized MAPSR (Chen et al., 2020) theoretically, which is combinatorially sample complex, the analysis is shown in §A.4. For a non-complete static graph with $k$ maximum number of degrees for $k \ll n$, which is more common in the real world, because a very far-away robot will not likely affect the targeted robot, the operation will be significantly decreased to $\mathcal{O}(k^2)$.

## 4.4 THE ESTIMATION FOR GAPSR MODEL COMPONENTS

As stated by Littman et al. (2001), a complete PSR model can build a recursive rule to update itself. In other words, given the $\mathbf{q}_{t,i,j}$, the GAPSR model can calculate $\mathbf{q}_{t+1,i,j}$ by incorporating a new observation, a so called *filtering* process in the dynamical system. We achieve this with

three models proposed by Hefny et al. (2018a): For any agent $i$, $\mathbf{p}_{t,i,j} = W_{i,j}(\mathbf{q}_{t,i,j})$, $\mathbf{q}_{t+1,i,j} = F_{i,j}(\mathbf{p}_{t,i,j}, o_{t,i}, a_{t,j})$, and $o_{t,i} = Z_{i,j}(\mathbf{q}_{t,i,j}, a_{t,j})$. The last model is used to predict the observation $o_{t,i}$ with action $a_{t,j}$ and PSR $\mathbf{q}_{t,i,j}$. Typically, $W_{i,j}, Z_{i,j}$ are learnable linear maps, and $F_{i,j}$ is non-linear and differentiable but known in advance. The $W_{i,j}, Z_{i,j}$ are learned from regression after we estimate $\mathbf{q}_{t,i,j}$ and $\mathbf{p}_{t,i,j}$, see §A.2.2 for details of estimation of $W_{i,j}, Z_{i,j}$.

## 5 DECISION-MAKING FRAMEWORK WITH GAPSR MODEL

Previous work on single-agent proposed an end-to-end training algorithm (Hefny et al., 2018a) for PSR model and policy learning. Here we design an algorithm for multi-agent settings and incorporate our GAPSR model containing the interactive graph component.

We propose an online learning algorithm to learn the GAPSR and agent policies simultaneously. Our algorithm is shown in §B.2. We use a diagram in §B.3 to illustrate this process.

Our algorithm has three components. Firstly, we estimate the GAPSR the model parameters $W_{i,j}$ and $Z_{i,j}$ (line 4-11). The agents use an existed $T$ length trajectory generated based on the iteration $k-1$'s policy (a random policy is used if it is the first time), and learn the GAPSR parameters under the given interactive graph by regressions with details introduced in 4.4.

Secondly, we use the just learned GAPSR to generate the predictive state representations $\mathbf{q}$ and $\mathbf{p}$ (line 13-28). In particular, by starting from the initialized GAPSR and executing the iteration $k-1$'s policy, the agents experience a $T$ length trajectory. The agents perform a series of extension, filtering, and prediction steps to generate the $\mathbf{q}_t$ and $\mathbf{p}_t$. Every agent also obtains action by executing its policy function that maps predictive states $\mathbf{q}_{t,i}$ to action $a_{t,i} \sim \pi_i^{k-1}(\mathbf{q}_{t,i})$. Agents then save trajectories (actions, observations, predictive states, and rewards) over the path.

Lastly, learning and planning via multi-agent actor-critic (line 30-34): agents use learned representations as the state and feed them into a MARL algorithm, e.g., MADDPG (Lowe et al., 2017). The MARL algorithm conducts the policy learning. Here we give two examples of GAPSR based MARL algorithms, which are implemented for our experiments. They follow the actor-critic framework. We first develop an algorithm under partially observable environments, where each agent has its own critic and actor independently, it is analogous to the independent actor-critic (IAC) (Foerster et al., 2018). The gradient of the policy is written as $\nabla_{\theta_i} J(\theta_i) = \mathbb{E}_{\tau_i \sim p(\tau_i|\Theta_i)}\left[\sum_{t=1}^{T} \nabla_{\theta_i} \log \pi_{\theta_i}(a_{t,i}|\mathbf{q}_{t,i})(r_i + \gamma V_{w_i}^{\pi_i}(\mathbf{q}_{t+1,i}) - V_{w_i}^{\pi_i}(\mathbf{q}_{t,i}))\right]$, and the independent critic is updated by minimizing the loss $\mathcal{L}_c = \mathbb{E}_{\tau_i \sim p(\tau_i|\Theta_i)}[(V_{w_i}^{\pi_i}(\mathbf{q}_i) - y_i)^2]$. Here the predictive states $\mathbf{q}_i$ generated by the GAPSR are considered as states to fit the value function. $\pi_\theta : \mathcal{Q} \times \mathcal{A} \to [0,1]$ is the stochastic policy that maps to the probability density of actions with parameter $\theta$. We call it GAPSR-1. As we know, IAC would not work well since the environment is not stationary under a multi-agent setting, the MADDPG by Lowe et al. (2017) solves a non-stationary environment by considering other agents actions; however, every agent needs a separate critic that has the global information. We used a centralized critic to minimize the loss $\mathcal{L}_c = \mathbb{E}_{\mathbf{q},r,o}\left[(Q_w^{\pi}(\mathbf{q}_1, \cdots, \mathbf{q}_n, a_1, \cdots, a_n) - y)^2\right]$. So our policy gradient is written as $\nabla_{\theta_i} J(\theta_i) = \mathbb{E}_{\mathbf{q},r,o}\left[\nabla_{\theta_i} \pi_{\theta_i}(\mathbf{q}_i) \nabla_{a_i} Q_w^{\pi}(\mathbf{q}_1, \cdots, \mathbf{q}_n, a_1, \cdots, a_n)|_{a_i=\pi_{\theta_i}(\mathbf{q}_i)}\right]$. To fully consider other agents' information, it uses the joint predictive states as input. Here by abuse of notation, we use $\pi_\theta : \mathcal{Q} \to \mathcal{A}$ to indicate the deterministic policy with parameter $\theta$. We call it GAPSR-2. We put our integration details in §B.4.

Our algorithm has the following characteristics. First, due to its decoupled structure, it is a general algorithm in the sense that planning can utilize any MARL algorithm for multi-agent MDP.

Second, our algorithm is an end-to-end framework, in the implementation, we build an additive loss function and fully differentiate it with respect to model parameters. In particular, we update the MARL and GAPSR model parameter $\Theta = \{\Theta_{\text{GAPSR}}, \Theta_{\text{MARL}}\}$ (line 32) by minimizing the following additive objective function:

$$\mathcal{L}(\Theta) = \alpha_1 \mathcal{L}_1(\Theta_{\text{MARL}}) + \alpha_2 \mathcal{L}_2(\Theta_{\text{GAPSR}}, \Theta_{\text{MARL}}), \tag{13}$$

$$\mathcal{L}_1(\Theta_{\text{MARL}}) = -J(\theta) + \mathcal{L}_c(w), \tag{14}$$

$$\mathcal{L}_2(\Theta_{\text{GAPSR}}, \Theta_{\text{MARL}}) = \sum_{i=1}^{n} E_{\tau \sim p(\tau|\Theta)}\left[\sum_j \left\| Z_{i,j}\left(F_{i,j}(W_{i,j}(\mathbf{q}_{t-1,i,j})) \otimes \phi_{t,j}^a\right) - \phi_{t,i}^o\right\|_2^2\right]. \tag{15}$$

Here the $\mathcal{L}_1(\Theta_{\text{MARL}})$ is the objective function for MARL, for example, if the actor-critic used, it will be the negative action value function with policy parameter $\theta$ plus the critic loss with parameter $w$, and $\Theta_{\text{MARL}} = (\theta, w)$. $\mathcal{L}_2(\Theta_{\text{GAPSR}}, \Theta_{\text{MARL}})$ is the MSE between prediction and actual observation. And $p(\tau|\Theta)$ is the distribution over trajectories induced by the policy and GAPSR. $\Theta_{\text{MAPSR}} = \{W_{i,j}, Z_{i,j}\}$ denotes GAPSR's parameters. The $\alpha_1$ and $\alpha_2$ are hyper-parameters to penalize differently on two losses.

Third, PSR, as a function of action, can bring agents' action information into the policy gradient; our interactive GAPSR even brings the effect of other agents' actions to the policy gradient as well.

## 6 EXPERIMENTS

**Environments.** We evaluate the performance of our GAPSR on a collection of MARL tasks under some OpenAI Gym MAMuJoco environments (de Witt et al., 2020), such as multi-agent swimmer, hopper, and ant. Each robotic agent is represented as a body graph, where vertices (joints) are connected by adjacent edges (body segments) as shown in Appendix Figure 5. Each agent controls its joints based on the local information observed. All tasks are learned under the partially observable environments by manually hiding some observations for each agent. The goals of the multi-agent systems are aligned with their corresponding single-agent ones. However, different from the single-agent system, the agents in the multi-agent system need to collaborate to reach their goals. For the interactive graph, for simplicity, we considered a complete static graph, where we assume every agent is connected with all other agents in the graph. We defer the details of the robotic agents to §C.1 and the details of the experimental setup to §C.2.

**Baselines and Evaluation.** We run 50 iterations for each experiment and collect $M = 100$ trajectories in every iteration with a maximum of 1000 steps in every trajectory. After each iteration, we compute the average return $R = \frac{1}{M} \sum_{i=1}^{n} \sum_{b=1}^{M} \sum_{t=1}^{T_b} r_{i,b}^t$ on a batch of $M$ trajectories, where, $T_b$ is the length of the $b^{th}$ trajectory. We repeat this process using ten

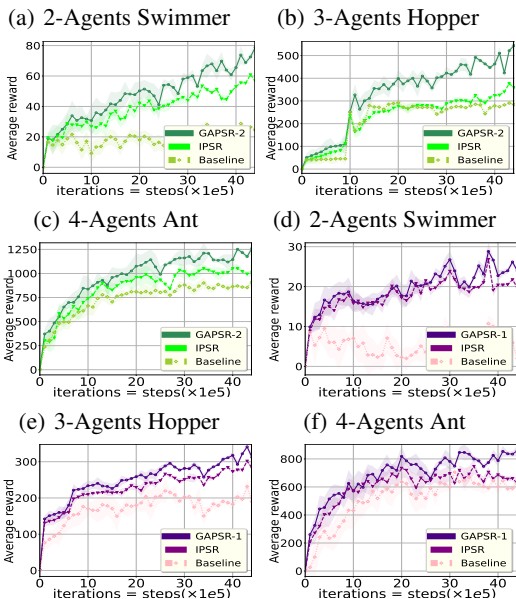

Figure 1: Performance of our method under MA-MuJoco partially observable environments, the ablation study– IPSR, in which we do not consider the agent interaction, and the baseline, which is not using GAPSR. (a)-(c):GAPSR-2 (It uses centralized critic, and uses gradients of value function with respect to policy parameter, its baseline is MADDPG), (d)-(f):GAPSR-1 (Its baseline is IAC). We run 10 times and the shaded area is the 95% confidence interval

different random seeds and report the average and a standard deviation. To verify the effectiveness of interactive graph, we introduce a baseline called Independent PSR learning (IPSR); in this model, we do not consider the graph, so we formulate $n$ independent single PSR without considering their interactions, which means there are no $\mathbf{q}_{i,j}$ any more but only $\mathbf{q}_i$. The architecture of GAPSR-1 and GAPSR-2 remains the same. To verify the advantage of PSR, we introduce another baseline (MARL) where we take out the GAPSR entirely, so it matches with the MARL run on a partially observable environment. For GAPSR-1, its MARL baseline is IAC (Foerster et al., 2018), and for GAPSR-2, is MADDPG (Lowe et al., 2017).

**Results And Discussion** Figure 1 illustrates the empirical average return vs. the number of interactions with the environment measured in time steps. Our GAPSR methods consistently outperform IPSR and get the highest rewards under partially observable environments, which justifies the representation power of the interactive graph to assist agents in learning in the non-stationary environment with limited observation when the MARL algorithm has a defect (IAC). Moreover, it can also boost the performance of the existing good MARL algorithm (MADDPG). To further verify the effect of learning the PSR part, we also plotted the predicted trajectory to verify the GAPSR's performance for predicting the observations in Figure 2. We plotted the predicted observations vs. actual observations in iterations $1$, and $40$, respectively, for GAPSR-2. We plotted a row $\times$

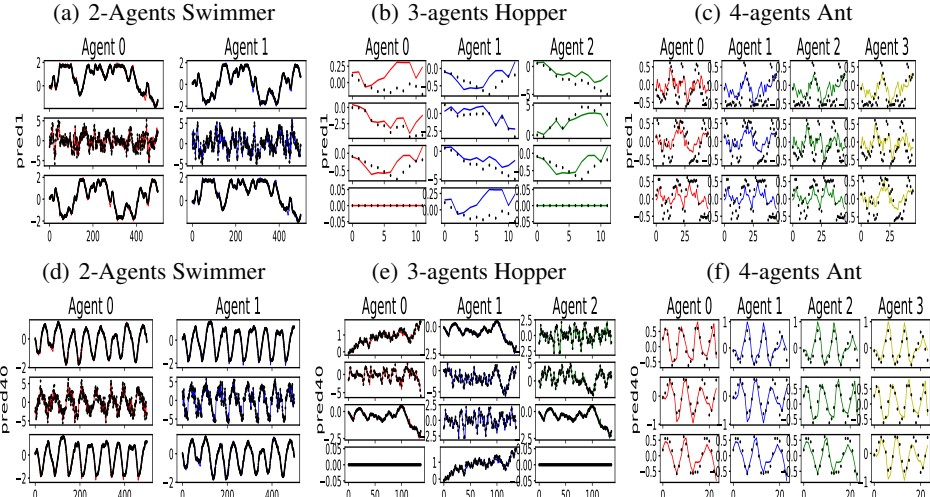

Figure 2: Predicted Trajectories (colored) vs Actual Observations (black). (a)-(c) First iteration; (d) - (f) Iteration 40. The $X$ axis represents the part of steps encountered for one trajectory under that iteration, and the $Y$ axis represents the numerical value of the observation, i.e. (a) has three rows to represent its three coordinates of its observation. We also provide iteration 10, 20 in §D.1

columns figure, with each row representing the observation feature and each column representing each agent. As we can see from the figures, the first iteration of the learning does not predict the actual observation very well; it has some mismatches. However, as learning progresses, the predictions get increasingly more accurate. Note that the actual trajectory is changing according to the current policy, and the current policy is optimized based on the further accurate learning of GAPSR.

To enrich our algorithm environment, we also test our algorithm (GAPSR-2) in multi-agent particle environments, using the benchmark by (Lowe et al., 2017), please check §D.2 for experiment details. We test our algorithm for large $n$ cases. In this environment, the agents can have cooperative goals such that all agents must maximize a shared return and conflicting competitive goals. We set up the environments where agents can only perform physical actions but not communication; however, to achieve the goals, agents need explicit communication about others' locations to achieve the best reward. These partially observable environments give us the motivation to test our method. We report the rewards in Figure 3. We see that GAPSR outperforms IPSR and baseline, in terms of the convergence speed and final attained rewards, with a different number of agents. The predictive states convey the information that can help communication between agents in limited communication and observation environments.

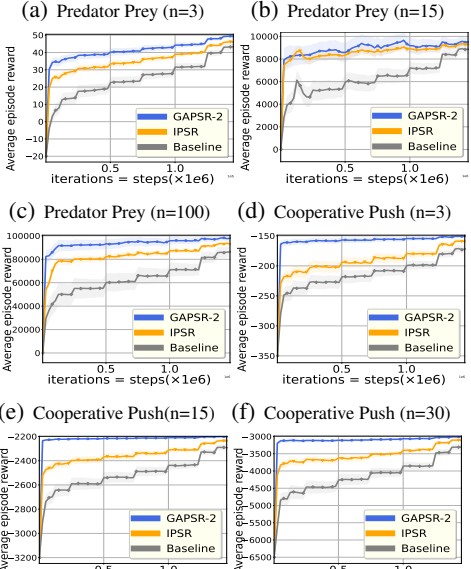

Figure 3: Performance of our method (GAPSR-2) under multi-agent particle partially observable environments. We use a different number of agents in predator-prey and cooperative push environments. We run ten times, and the shaded area is the 95% confidence interval.

## 7 CONCLUSION

We propose a GAPSR model, extending ideas from single-agent predictive state representations to a multi-agent scenario, during the process, we introduce the dynamic interactive graph to model agents' interactions. Furthermore, we provide the theoretical guarantees of the GAPSR model. Finally, a learning algorithm that supports gradient-based deep MARL methods is developed. Our method provides a model-based MARL framework under a partially observable environment. The experiments proved that our model assumption is valid by observing the highest return while reducing the observations' prediction error over trajectories.

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

# A    METHOD DETAILS

## A.1    MATHEMATICAL MODEL OF PSR

Here, we give the mathematical model for PSR. As in Section 3, we let $\tau_t := \{(o_l, a_l)\}_{l=t}^{t+k-1}$ as a test (a sequence of action-observation pairs) at time $t$, then we have a subset $\tau_t^o := \{o_l\}_{l=t}^{t+k-1}$ for observations, and $\tau_t^a := \{a_l\}_{l=t}^{t+k-1}$ for actions. And we let $h_t := \{(o_l, a_l)\}_{l=1}^{t-1}$ as the history at time $t$. A test is executed at time $t$ if we intervene to select the sequence of actions specified by the test. It is said to succeed at time $t$ if it is executed and the sequence of observations in the test matches the observations by the system (Boots et al., 2013).

The prediction of a length-$k$ test $\tau_t$ is defined as the probability distribution that the test succeed at time $t$, given history $h_t$: $\mathbf{Pr}(\tau_t^o|\tau_t^a, h_t) = \frac{\mathbf{Pr}(\tau_t^o, \tau_t^a|h_t)}{\mathbf{Pr}(\tau_t^a|h_t)}$

Let $T = \{\tau_z\}, z \in N$ is a finite set of core tests. And we use $T^o = \{\tau_1^o, \cdots, \tau_{|T|}^o\}$ and $T^a = \{\tau_1^a, \cdots, \tau_{|T|}^a\}$ to indicate its observation and action parts.

### A.1.1    LINEAR PSR

A linear PSR is the prediction vector

$$\mathbf{q}_t := \left[\mathbf{Pr}(\tau_1^o|\tau_1^a, h_t), \ldots \mathbf{Pr}(\tau_{|T|}^o|\tau_{|T|}^a, h_t)\right]^\top \tag{16}$$

that contains the probabilities of success of the tests in T, if and only if for any test $\tau$

$$\mathbf{Pr}(\tau^o|\tau^a, h_t) = f_\tau(\mathbf{q}_t), \tag{17}$$

where $f_\tau : [0, 1]^{|T|} \mapsto [0, 1]$ is a linear projection function. For simplicity of notation, we use the same notations for $\mathbf{q}_t \mathbf{p}_t, W, F$, which only follows the meaning defined in this section.

It means that knowing the probabilities for the tests in T is sufficient for computing the probabilities for all other tests in the system. The prediction vector is a sufficient statistic for the system at time $t$, so we call it as **state** for the **PSR** at time $t$. Therefore, PSR can represent state by using a vector of predictions of fully observable quantities (tests) conditioned on past events (histories). The prediction function $f_\tau$ is linear and has one-to-one relationship to a test $\tau$ such that $f_\tau(\mathbf{q}_t) = f_\tau^\top \mathbf{q}_t \quad \forall t$. The linear PSR can still represent systems with nonlinear dynamics.

To maintain predictions in T, we need to update the state $\mathbf{q}_t$. To do that, we predict the success of any core test $\tau_z$ prepended by a new action $a$ and observation $o$ at time $t + k - 1$, which we call $oa\tau_z$.

Based on Bayes rule, we have

$$\mathbf{Pr}(\tau_{z,t+1}^o|\tau_{z,t+1}^a, h_{t+1} = (h_t, o, a))) = \frac{\mathbf{Pr}(o\tau_{z,t+1}^o|a\tau_{z,t+1}^a, h_t)}{\mathbf{Pr}(o|a, h_t)} = \frac{f_{ao\tau_z}^\top \mathbf{q}_t}{f_{ao}^\top \mathbf{q}_t}, \tag{18}$$

and $f_{ao\tau_z}, f_{ao} \in \mathbb{R}^{|T|}$ are linear operators such that $(\forall \tau_z \in T, \forall a \in \mathcal{A}, \forall o \in \mathcal{O})$. Then let $F_{ao\tau}$ be the matrix with its columns as $f_{ao\tau_z}$ for all $\tau_z \in T$. Then the updated state in PSR is obtained by

$$\mathbf{q}_{t+1} = \frac{F_{ao\tau}^\top \mathbf{q}_t}{f_{ao}^\top \mathbf{q}_t}. \tag{19}$$

Given the initial prediction vector $\mathbf{q}_1$, the PSR can update with equation 19. This recursive application of Bayes rule to a belief state is called the Bayes filter. Now we have seen the extended prediction vector, so we define **extended predictive state** as:

$$\mathbf{p}_t := \left[\mathbf{Pr}(a\tau_1^o|a\tau_1^a, h_t), \ldots \mathbf{Pr}(a\tau_{|T|}^o|a\tau_{|T|}^a, h_t)\right]^\top. \tag{20}$$

Clearly, we can see there exists a linear extension operator $W$ such that $\mathbf{p}_t = W(\mathbf{q}_t)$. And given the new observation, we can have a Bayes filter $F$ such that $\mathbf{q}_{t+1} = F(\mathbf{p}_t, o, a)$. We also introduced these operators in section 4.4.

### A.1.2    REPRESENTATION OF STATE AS CONDITIONAL EXPECTATION OF SUFFICIENT STATISTICS

Instead of learning the distribution, here we recover the idea of representation of states as a conditional expectation of sufficient statistics and using the supervised learning method to learn (Hefny et al.,

2015). Let history $h_t^o$ is a sequence of observation that starts from the beginning of time and ends at time $t - 1$.

We define the belief state $b_t = \mathbf{Pr}(s_t|h_t^o)$, where $s_t$ is the current state of the world. $b_{t+1} = \mathbf{Pr}(s_{t+1}|h_{t+1}^o)$ is the next time belief state. We call $b_t$ as "belief state", which represents the probability distribution over state space, also represents the knowledge and uncertainty about the true state of the system. In dynamic system, the task of getting the updated $b_{t+1}$ with given $b_t$ and new observation $o_t$ is called filtering. The task of estimating the $\mathbf{Pr}(s_{t+1}|h_t^o)$ with given current $b_t$ without incorporating any new observation is called one-step prediction.

Instead maintaining a belief $b_t$ over states, spectral algorithms try to recover observable operators that can be used to perform filtering and prediction directly, by maintaining the expected value of a sufficient statistic of future observations.

Let a test $\tau_t^o$ is a length $k$ sequence of observation defined before. As the recursive Bayes rule holds, we can get the next time prediction vector $\mathbf{Pr}(\tau_{t+1}^o|h_{t+1}^o)$ using the new observation $o_{t+k}$ and the current prediction $\mathbf{Pr}(\tau_t^o|h_t^o)$, we also define an extended prediction vector $\mathbf{Pr}(\tau_{t+1}^o|h_t^o)$. These prediction vectors characterize the state of the system and they can be estimated by observable quantities. Given $\mathbf{Pr}(\tau_{t+1}^o|h_t^o)$, filtering becomes the task of getting the updated prediction vector $\mathbf{Pr}(\tau_{t+1}^o|h_{t+1}^o)$, "conditioning" on $o_t$. One-step prediction becomes getting the $\mathbf{Pr}(\tau_{t+1}^o|h_t^o)$, "marginalizing" over $o_t$.

Therefore, the spectral algorithms avoid explicitly estimating the latent state or the initial, transition, or observation distributions. We let $\mathbf{q}_t = \mathbb{E}[\nu_t|h_t^o]$, where $\nu_t$ a vector of features that determines the distribution of future observations $\mathbf{Pr}(\tau_t^o|h_t^o)$. For simplicity of notation, we use the same notations for $\mathbf{q}_t, \mathbf{p}_t, W, F$. Let $\mathbf{p}_t = \mathbb{E}[\varphi_t|h_t^o]$, where $\varphi_t$ is a vector of features that determines the distribution of observations $\mathbf{Pr}(\tau_{t+1}^o|h_t^o)$. We call $\mathbf{q}_t$ is the transformed predictive state. So $\mathbf{q}_{t+1} = \mathbb{E}[\nu_{t+1}|h_{t+1}^o]$, is the updated predictive state, and $\mathbf{p}_t$ is the extended predictive state. Let $\mathrm{h}_t^o$ be the feature vector of history $h_t^o$.

In Hidden Markov Models and Kalman filters, the extended state $\mathbf{p}_t$ is linearly related to the predictive state $\mathbf{q}_t$. Which is $\mathbf{p}_t = W\mathbf{q}_t$. Estimation of the $W$ can be done using linear regression with samples $\nu_t$ and $\varphi_t$, however, due to the overlap between observation windows, the noise terms on $\nu_t$ and $\varphi_t$ are correlated, which will cause biased estimate. The instrumental regression (Pearl et al., 2000; Stock et al., 2012) is employed. $\mathrm{h}_t^o$ is a instrumental variable that do not overlap with sequence $\{l\}_{l=t}^{l=t+k-1}$ and $\{l\}_{l=t}^{l=t+k}$. The correlation $\mathrm{corr}(\mathrm{h}_t^o, \mathrm{e}(\varphi_t)) = 0$ and $\mathrm{corr}(\mathrm{h}_t^o, \mathrm{e}(\nu_t)) = 0$, where e is a measure of error. By taking the conditional expectation of $\mathbf{p}_t = W\mathbf{q}_t$ given $\mathrm{h}_t^o$, we have

$$\mathbb{E}[\mathbf{p}_t|\mathrm{h}_t^o] = \mathbb{E}[W\mathbf{q}_t|\mathrm{h}_t^o], \tag{21}$$
$$\mathbb{E}[\mathbb{E}[\varphi_t|h_t^o]|\mathrm{h}_t^o] = W\mathbb{E}[\mathbb{E}[\nu_t|h_t^o]|\mathrm{h}_t^o],$$
$$\mathbb{E}[\varphi_t|\mathrm{h}_t^o] = W\mathbb{E}[\mathbf{q}_t|\mathrm{h}_t^o].$$

Based on the above relationship, we first estimate the $\mathbb{E}[\varphi_t|\mathrm{h}_t^o]$ and $\mathbb{E}[\mathbf{q}_t|\mathrm{h}_t^o]$ by sample $\mathrm{h}_t^o$, $\nu_t$, and $\varphi_t$, we then use the estimates to compute $W$. So if we start with $\mathbf{q}_1$, we can compute $\mathbf{p}_1 = W\mathbf{q}_1$, and get the $\mathbf{q}_{t+1} = F(\mathbf{p}_1, o_1)$, where $F$ is the Bayes filter to update the state.

## A.2 LEARNING FOR GAPSR IN DETAILS

Here we introduce the details about learning GAPSR, which supplements the section 4.4 and 4.1.1.

### A.2.1 LEARNING FOR $\widehat{\mathbf{q}}_{t,i,j}$ AND $\widehat{\mathbf{p}}_{t,i,j}$

To calculate $\widehat{\mathbf{q}}_{t,i,j}$, we introduce two sets of linear operators $T_{i,j}$ and $U_{i,j}$, such that $T_{i,j}(\psi_{t,i,j}^h) \approx C_{\psi_{t,i}^o \psi_{t,j}^a|\psi_{t,i,j}^h}$ and $U_{i,j}(\psi_{t,ij}^h) \approx C_{\psi_{t,j}^a \psi_{t,j}^a|\psi_{t,i,j}^h}$. We estimate them by using two ridge regressions:

$$\arg\min_{T_{i,j}} \sum_{t=1}^{T} \mathcal{L}(T_{i,j}(\psi_{t,i,j}^h)), \psi_{t,i}^o \otimes \psi_{t,j}^a) + R(T_{i,j}), \tag{22}$$

$$\arg\min_{U_{i,j}} \sum_{t=1}^{T} \mathcal{L}(U_{i,j}(\psi_{t,i,j}^h), \psi_{t,j}^a \otimes \psi_{t,j}^a) + R(U_{i,j}). \tag{23}$$

$\mathcal{L}$ represents the ridge regression loss, and $R$ represents the regularizer. After learning $T_{i,j}$ and $U_{i,j}$, we can get the estimate of $C_{\psi^o_{t,i}\psi^a_{t,j}|\psi^h_{t,i,j}}$ and $C_{\psi^a_{t,j}\psi^a_{t,j}|\psi^h_{t,i,j}}$. Then we get the $\widehat{\mathbf{q}}_{t,i,j}$ by equation 7. Similarly, we use extended features to obtain $\widehat{\mathbf{p}}_{t,i,j}$.

### A.2.2 Extension, Filtering and Prediction Functions

**Filtering.** To obtain $\widehat{\mathbf{q}}_{t+1,i,j}$ from $\widehat{\mathbf{q}}_{t,i,j}$, $\widehat{\mathbf{p}}_{t,i,j}$, we use filtering. We denote $F_{i,j}$ as the filtering function. We describe the filtering process as below. From $\{o_i\}_{i=1}^n$ and $\{a_j\}_{j=1}^n$, we obtain the embedding $\{\phi^o_{t,i}\}_{i=1}^n$ and $\{\phi^a_{t,j}\}_{j=1}^n$. We then compute the observation covariance

$$C_{o_{t,i},o_{t,i}|h_{t,i,j},a_{t,j}} = M_{\phi^o_{t,i}\otimes\phi^o_{t,i}|\phi^a_{t,j};\psi^h_{t,i,j}}\phi^a_{t,j}. \tag{24}$$

We then multiply the extended state by inverse observation covariance to change predicting $\phi^o_{t,i}$ into conditioning on $\phi^o_{t,i}$.

$$M_{\psi^o_{t+1,i}|\psi^a_{t+1,j},\phi^o_{t,i},\phi^a_{t,j};\psi^h_{t,i,j}} = M_{\psi^o_{t+1,i}\otimes\phi^o_{t,i}|\psi^a_{t+1,j},\phi^a_{t,j};\psi^h_{t,i,j}} \times_{\phi^o_{t,i}} (C_{o_{t,i},o_{t,i}|h_{t,i,j},a_{t,j}} + \lambda I)^{-1}. \tag{25}$$

$\times$ here is to denote $n-mode$ (matrix or vector) product, $\times_{\phi^o_{t,i}}$ means multiplying the tensor by a matrix (or vector) in mode $\phi^o_{t,i}$.

We condition on $\phi^o_{t,i}$ and $\phi^a_{t,j}$ to obtain shifted state.

$$\mathbf{q}_{t+1,i,j} := M_{\psi^o_{t+1,i}|\psi^a_{t+1,j};\phi^o_{t,i},\phi^a_{t,j},\psi^h_{t,i}} = M_{\psi^o_{t+1,i}|\psi^a_{t+1,j},\phi^o_{t,i},\phi^a_{t,j};\psi^h_{t,i}} \times_{\phi^o_{t,i}} \phi^o_{t,i} \times_{\phi^a_{t,j}} \phi^a_{t,j}. \tag{26}$$

Based on the updating rule, $\mathbf{q}_{t+1,i,j} = F_{i,j}(\mathbf{p}_{t,i}, o_{t,i}, a_{t,j})$, and $\mathbf{p}_{t,i} = W_{i,j}(\mathbf{q}_{t,i,j})$, we write the filtering equation.

$$\mathbf{q}_{t+1,i,j} = F_{i,j}(\mathbf{p}_{t,i}, o_{t,i}, a_{t,j}) \tag{27}$$

$$:= M_{\psi^o_{t+1,i}\otimes\phi^o_{t,i}|\psi^a_{t+1,j},\phi^a_{t,j};\psi^h_{t,i,j}} \times_{\phi^o_{t,i}} (M_{\phi^o_{t,i}\otimes\phi^o_{t,i}|\phi^a_{t,j};\psi^h_{t,i,j}}\phi^a_{t,j} + \lambda I)^{-1} \times_{\phi^o_{t,i}} \phi^o_{t,i} \times_{\phi^a_{t,j}} \phi^a_{t,j},$$

where $\mathbf{p}_{t,i} := M_{\psi^o_{t+1,i}\otimes\phi^o_{t,i}|\psi^a_{t+1,j},\phi^a_{t,j};\psi^h_{t,i,j}}$. $F_{i,j}$ usually is known because it is obtained through the above calculation using known quantities, however, $W_{i,j}$, $Z_{i,j}$ must have to be learned by using regressions.

**Extension.** The $W_{i,j}$ can be learned by kernel regression if we know the $\widehat{\mathbf{q}}_{t,i,j}$ and $\widehat{\mathbf{p}}_{t,i,j}$. Previous work (Hefny et al., 2018a) demonstrated the kernel regression model for learning single-agent PSR, here we extend to the GAPSR. We set the model parameter $W_{i,j}$.

We optimized a ridge regression problem for $W_{i,j}$,

$$\underset{W_{i,j}}{\arg\min} \sum_{t=1}^T \mathcal{L}(W_{i,j}(\widehat{\mathbf{q}}_{t,i,j}), \widehat{\mathbf{p}}_{t,i,j}) + R(W_{i,j}). \tag{28}$$

**Prediction.** We can also get the prediction about the next one time observation $o_t$ by the regression function such as:

$$\widehat{o}_{t,i} := \mathbb{E}(o_{t,i}|\mathbf{q}_{t,i,j}, a_{t,j}) = Z_{i,j}(\mathbf{q}_{t,i,j} \otimes \phi^a_{t,j}). \tag{29}$$

We solve the prediction regression function $Z_{i,j}$ by another ridge regression:

$$\underset{Z_{i,j}}{\arg\min} \sum_{t=1}^T \mathcal{L}(Z_{i,j}(\widehat{\mathbf{q}}_{t,i,j} \otimes \phi^a_{t,j}), \phi^o_{t,i}) + R(Z_i). \tag{30}$$

### A.3 Tensor Decomposition of MAPSR

**Existed Formulation.** (Chen et al., 2020) use a $n + 1$ multi-dimensional tensor called system dynamics tensor $\mathcal{D} \in \mathbb{R}^{|\mathcal{T}_1|\times\cdots\times|\mathcal{T}_n|\times|\mathcal{H}|}$ to represent the system dynamics of the MAPSR, with $|\mathcal{T}_1|$ representing the cardinality of the tests for agent 1, $n$ denoting the number of agents, and the $|\mathcal{H}|$ being the cardinality of the joint history. Each element of the tensor is a probability of a joint test given joint histories. Given the system dynamics tensor, finding the latent predictive state can be transferred into finding a minimal linearly independent set from the system dynamics tensor, and it can be solved by spectral method such as tensor decomposition.

As mentioned earlier, this formulation can not satisfy our needs. In **Proposition 1** of Appendix A.4, we also give the sample complexity analysis such that the sample size needed to formulate $\mathcal{D}$ scales exponentially with the length of tests and number of agents.

Here we give a summary of tensor decomposition of their method. Given a system dynamic tensor $\mathcal{D}$ such that:

$$\mathcal{D} \approx [\lambda; D^1, \cdots, D^n, F] = \sum_{r=1}^{R} \lambda_r D_r^1 \circ \cdots \circ D_r^n F_r. \tag{31}$$

Here $\circ$ is the outer product. $D^1, \cdots, D^n, F$ are matrices. The factor matrices $D^1$, $D^n$, $F$ consist of the vectors, i.e., $D^1 = [D_{:1}^1 D_{:2}^1 \cdots D_{:R}^1] \in \mathbb{R}^{|\mathcal{T}_1| \times R}$. A colon is used to indicate all elements of a mode, thus, the $R^{th}$ column of $D^1$ is denoted by $D_{:R}^1$. For any $i_1 \in \{1, \ldots, |\mathcal{T}_1|\}$, $i_n \in \{1, \ldots, |\mathcal{T}_n|\}$, and $k \in \{1, \ldots, |\mathcal{H}|\}$, after the decomposition, we get

$$\mathcal{D}_{i_1, \cdots, i_n, k} = \sum_{r=1}^{R} \lambda_r D_{i_1 r}^1 \cdots D_{i_n r}^n F_{kr}, \tag{32}$$

where $\{i_1, \cdots, i_n, k\}$ are index corresponding to the specific dimension of the $\mathcal{D}$. The last dimension of the tensor $\mathcal{D}$ is compressed in a matrix $F$, and its row vector $x_k = [x_k(1) \ldots x_k(R)] \in \mathcal{R}^{1 \times R}$, $k \in \{1, \ldots, |\mathcal{H}|\}$ is a summary of joint history and can be considered as a compressed version of the system predictive state vector $p(\mathbf{Q}|\mathbf{h}_k)$, the joint history $\mathbf{h}_k \in \mathcal{H}(k \in [1, |\mathcal{H}|])$ at time step $s = |\mathbf{h}_k|$, where $\mathbf{Q}$ is the core joint test set. The whole fibers listed in the set $\mathbf{Q}$ form a basis of the space spanned by the mode-(n+1) fibers of tensor $\mathcal{D}$. Thus, by constructing the vector $m = (\lambda * D_{i_1:}^1 * \cdots D_{i_n:}^n)^T$, where $*$ is Hadamard product. $D_{i_1:}^1$ denotes the $i_1$-th row vector of $D^1$. Then we could rewrite the previous equation as

$$\mathcal{D}_{i_1, \cdots, i_n, k} = \sum_{r=1}^{R} m(r) x_k(r) = x_k(\lambda * D_{i_1:}^1 * \cdots D_{i_n:}^n)^T = x_k m. \tag{33}$$

$m(r)$ is a scalar that $m(r) = \lambda_r D_{i_1 r}^1 \cdots D_{i_n r}^n$. And $x_k$ is the system state vector and $m$ is the prediction parameter, both of them are obtained by the tensor decomposition.

## A.4 ANALYSIS OF SAMPLE COMPLEXITY FOR FORMULATING THE SYSTEM DYNAMIC TENSOR

The paper (Chen et al., 2020) does not analyze the sample complexity to construct $\mathcal{D}$, which is a $n + 1$ multi-dimensional tensor $\mathcal{D} \in \mathbb{R}^{|\mathcal{T}_1| \times \cdots \times |\mathcal{T}_n| \times |\mathcal{H}|}$. We give this analysis. The $\mathcal{D}_{i_1, \cdots, i_n, k} := \mathbf{Pr}(t_{1i_1}, \cdots, t_{ni_n}|\mathbf{h}_k)$ is an element of that tensor $\mathcal{D}$ such that $t_{1i_1}$ is the $i_1$-th test of agent 1, similarly, the $t_{ni_n}$ is the $i_n$-th test of agent $n$, and $\mathbf{h}_k$ is the joint history.

**Proposition 1.** *In a $n$-agents system, assume every agent has the same observation and action space $|\mathcal{O}|$, $|\mathcal{A}|$, for a length-$k$ test, to formulate a complete system dynamics tensor $\mathcal{D}$ defined in equation 32. Assume each entry of the tensor needs $S$ samples to give a sufficient estimation using Monte-Carlo roll-out method, then the total sample size is at least $(|\mathcal{O}||\mathcal{A}|)^{kn}S$, if the agents are homogeneous, in other words, they are permutation invariant such that identity does not matter, then the total sample size is $\binom{|\mathcal{O}||\mathcal{A}|^k + n - 1}{n}S$.*

*Proof.* At one time step, for any agent, it has $|\mathcal{O}||\mathcal{A}|$ different combinations for the joint test, then for a length of $k$ tests, it follows that $(|\mathcal{O}||\mathcal{A}|)^k$ number of different choices. Then the tensor $\mathcal{D}$ would need $(|\mathcal{O}||\mathcal{A}|)^{kn}$ elements to cover all the possible length $k$ tests. So the total sample size is $(|\mathcal{O}||\mathcal{A}|)^{kn}S$. If the agents are homogeneous, then the ordering does not matter, for each agent, we have $(|\mathcal{O}||\mathcal{A}|)^k$ number of different choices for length $k$ test, so the total choices for $n$ agents are $\binom{|\mathcal{O}||\mathcal{A}|^k + n - 1}{n}$, then we need $\binom{|\mathcal{O}||\mathcal{A}|^k + n - 1}{n}S$ samples. $\qquad \square$

The sample complexity is exponentially scaled with the number of agents and length of tests.

## B ALGORITHM AND INTEGRATING WITH MARL METHOD

We first introduce some backgrounds on two multi-agent frameworks: Multi-agent Markov Decision Process (MMDP) and Multi-agent Partially observable Markov Decision Process (MPOMDP) since the algorithms in our paper are developed based on the framework of MPOMDP.

### B.1 MULTI-AGENT MDP AND MULTI-AGENT POMDP MODEL

#### B.1.1 MMDP

MMDP model is a tuple $\left(\mathcal{S}, N, \{\mathcal{O}_i\}_{i \in [n]} \{\mathcal{A}_i\}_{i \in [n]}, T, R\right)$, where $\mathcal{S}$ and $N$ are finite sets of states and agents. $\mathcal{A}_i$ is a finite set of actions available to agent $i$; $T : \mathcal{S} \times \mathcal{A}_1 \times \cdots \times \mathcal{A}_n \times \mathcal{S} \to [0, 1]$ is a transition function; and $R : \mathcal{S} \to R$ is the reward function. Each agent $i$ obtains reward as function of the state and agent's action $r_i : \mathcal{S} \times \mathcal{A}_i \to R$, and receives a private observation from the state by the observation channel $o_i : \mathcal{S} \to \mathcal{O}_i$. The state has distribution $d : \mathcal{S} \to [0, 1]$. Each agent aims to maximize its own total expected return $r_i = \sum_{t=0}^{T} \gamma^t r_i^t$ where $\gamma$ is a discounted factor and $T$ is the time horizon. In a shared reward situation, there is a team reward function $r : \mathcal{S} \times \mathcal{A}_i \cdots \times \mathcal{A}_n \to R$, agents aim to maximize one shared total expected return $r = \sum_{t=0}^{T} \gamma^t r^t$.

#### B.1.2 MPOMDP

A MPOMDP model is a tuple $\left(\mathcal{S}, N, \{\mathcal{O}_i\}_{i \in [n]} \{\mathcal{A}_i\}_{i \in [n]}, \{\Omega_i\}_{i \in [n]}, T, R\right)$, where $(\mathcal{S}, \mathcal{A}_i, T_i, \mathcal{O}_i, \Omega_i, R)$ describe a single-agent POMDP. $\mathcal{O}_i$ is the set of observations the agent $i$ can make. $\Omega_i : \mathcal{S} \times \mathcal{A}_i \times \mathcal{O}_i \to [0, 1]$ is the agent's observation channel function, which specifies probabilities of observations given agent's actions and resulting states. $(\mathcal{S}, \mathcal{A}_i, T_i, R_i)$ describes a single agent MDP; and each agent $i$ obtaines reward as function of the state and agetn's action $r_i : \mathcal{S} \times \mathcal{A}_i \to R$. Each agent aims to maximize its own total expected return $r_i = \sum_{t=0}^{T} \gamma^t r_i^t$ where $\gamma$ is a discounted factor and $T$ is the time horizon. In POMDP, an agent's belief about the sate is represented as probability distribution over $\mathcal{S}$. The agent has prior belief $b_{0,i}$ The agent's current belief, $b_{t,i}$ over $\mathcal{S}$, is continuously revised based on new observations and expected results of performed actions. The belief update takes into account changes in initial belief, $b_{t-1,i}$, due to action $a_{t,i}$, executed at time $t - 1$, and the new observation, $o_{t,i}$. The new time belief state can be obtained from basic probability theory as follows: $b_i(s_t) = \beta \Omega_i(o_{t,i}, s_t, a_{t-1,i}) \sum_{s_{t-1} \in S} b_{t-1,i}(s_{t-1}) T(s_t, a_{t,i}, s_{t-1})$, where $\beta$ is the normalizing factor.

### B.2 ALGORITHM: GAPSR

We give the details of GAPSR in Algorithm 1.

### B.3 GAPSR IN DIAGRAM

We also use a diagram (Fig 4) to depict the algorithm. The algorithm runs $k$ iterations; each iteration first uses the policy to roll out data and uses the regressions to obtain the PSR parameters. Then it uses the PSR as the input of policy to generate action and using the current PSR parameters to update the predictive state. At the end of this phase, it updates both the PSR and policy parameters. The policy parameters and PSR parameters $W_i$ and $Z_i$ are parameterized by the neural network, section C.3 has the network architectures. The loss is a composite loss that includes loss from actor-critic and the loss from the predictive state representation. The next iteration will re-learn the PSR parameters using the newly generated data based on the current policy obtained from the previous iteration; then, it does a soft update to update the PSR parameters with ones obtained at the previous iteration.

### B.4 INTEGRATING TWO COMMON MARL ALGORITHMS INTO GAPSR MODEL

Here we provide a brief intuitive introduction about how we connect existed MARL algorithms with GAPSR. Please look at Fig 4 for demonstration. $\psi_{t,i}^o$, $\psi_{t,j}^a$, and $\psi_{t,i,j}^h$ are embedding of $(o_{t:t+k-1,i}, a_{t:t+k-1,j})$, and $(o_{1:t-1,i}, a_{1:t-1,j})$. We also have embedding for extended part, labeled as $\xi$. $\mathbf{q}_{t,i,j}$ and $\mathbf{p}_{t,i,j}$ are estimated by regression using the embedding vectors. $G$ represents the given graph, the estimation of $\mathbf{q}_{t,i}$ and $\mathbf{p}_{t,i}$ are based on graph $G$ and equation 5, equation 9 and equation 10 for reference. Please also go to section 4 for detailed description. The policy network uses the predictive state as input to return the action or its distribution. The agent takes the action to get the observation. The filter $F_{i,j}$ takes predictive state, action, and observation as inputs to get the next predictive state. We present a centralized critic and give the description in paragraph B.4.2. The loss is composed into two parts, we give a detailed explanation in section 5. Please also go to section 5 and Algorithm B.2 for more details about the framework.

---

**Algorithm 1** GAPSRL

---

1: **Input:** Learning rate $\eta$, a graph $G$, a static complete graph $G_c$ or static non-compete graph $G_s$ or dynamic graph $G_t$

2: Initialize MARL Policy $\Theta_{MARL}$ randomly

3: **for** $k = 1, 2, 3, \cdots$ iterations **do**

4:     **GAPSR Model Estimation Phase**

5:     Sample $b = 1, 2, 3, \cdots M$ batch of initial trajectories: $\{(o_t^b, a_t^b)\}_{b=1}^M$ from existed policy obtained from previous iteration $k-1$: $\{\pi_i^{k-1}\}_{i=1}^n$

6:     Let $\pi_i^k = \pi_i^{k-1}$ if available or the initial policy

7:     Given $G$, calculate $\widehat{\mathbf{q}}_{t,i,j}$ and $\widehat{\mathbf{p}}_{t,i,j}$, and obtain the initial $W_{i,j}, Z_{i,j}, F_{i,j}$:

8:       (1) Regression $\mathbf{q}_{t,i,j} = T_{i,j} \circ U_{i,j}(h_{t,i,j})$ to get the $\widehat{\mathbf{q}}_{t,i,j}$ and $\widehat{\mathbf{p}}_{t,i,j}$

9:       (2) Given the $\widehat{\mathbf{q}}_{t,i,j}, \widehat{\mathbf{p}}_{t,i,j}$, compute $W_{i,j}, Z_{i,j}, F_{i,j}$

10:       (3) Get initial $\mathbf{q}_{1,i,j}$ and $\mathbf{p}_{1,i,j}$ by using the $T_{i,j}$ and $U_{i,j}$ with the early window of observations.

11:       (4) Given the graph $G$, using equation 5, 9, or 10 to obtain $\mathbf{q}_{1,i}, \mathbf{p}_{1,i}$

12:

13:     **Generation of Predictive Representations**

14:     Initialize GAPSR parameters $\Theta_{GAPSR} = \{W_{i,j}, Z_{i,j}\}$ from GAPSR Model Estimation Phase and previous iteration by a soft-update:
$$W_{i,j} = \beta W_{i,j} + (1-\beta)W_{i,j}^{k-1}, Z_{i,j} = \beta Z_{i,j} + (1-\beta)Z_{i,j}^{k-1}, F_{i,j} = F_{i,j}$$

15:     **for** $b = 1, 2, 3, \cdots M$ batch of trajectories from $\{\pi_i^{k-1}\}_{i=1}^n$ **do**

16:       Reset episode: $a_1^b, o_1^b$ for all agents

17:       **for** $t = 1, 2, \cdots T$ roll-in in each trajectory **do**

18:         **for** Each agent $i$ **do**

19:           Get observation $o_{t,i}^b$ and reward $r_{t,i}^b$ and its *neighbor's* action $a_{t,j}^b$

20:           Extension $\mathbf{p}_{t,i,j}^b = W_{i,j}(\mathbf{q}_{t,i,j}^b)$

21:           Filtering $\mathbf{q}_{t+1,i,j}^b = F_{i,j}(\mathbf{q}_{t,i,j}^b, a_{t,j}^b, o_{t,i}^b, W_{i,j})$

22:           Predict $\widehat{o}_{t,i}^b = Z_{i,j}(\mathbf{q}_{t,i,j}^b, a_{t,j}^b)$

23:           Given the graph $G$, obtain $\mathbf{q}_{t+1,i}^b$

24:           Execute $a_{t+1,i}^b \sim \pi_i^{k-1}(\mathbf{q}_{t+1,i}^b)$

25:           Collect $o_{t,i}^b, \widehat{o}_{t,i}^b, a_{t,i}^b, r_{t,i}^b, \mathbf{q}_{t,i}^b$

26:         **end for**

27:       **end for**

28:     **end for**

29:

30:     **Learning Multi-agent Actor-Critic**

31:     Update $\Theta$ using $D = \{\{\{o_{t,i}^b, \widehat{o}_{t,i}^b, a_{t,i}^b, r_{t,i}^b, \mathbf{q}_{t,i}^b, \mathbf{q}_{t,i,j}^b\}_{i=1}^n\}_{t=1}^T\}_{b=1}^M$:
$\Theta \leftarrow \text{Update}(\Theta^{k-1}, D, \eta)$ as in Equation (13):

32:     $\mathcal{L}(\Theta) = \alpha_1 \mathcal{L}_1(\Theta_{\text{MARL}}) + \alpha_2 \mathcal{L}_2(\Theta_{\text{GAPSR}}, \Theta_{\text{MARL}})$
$\mathcal{L}_1(\Theta_{\text{MARL}}) = -J(\theta) + \mathcal{L}_c(w)$
$\mathcal{L}_c = \mathbb{E}_{\mathbf{q},r,o}\left[(Q_w^\pi(\mathbf{q}_1, \cdots, \mathbf{q}_n, a_1, \cdots, a_n) - y)^2\right]$
$\nabla_{\theta_i} J(\theta_i) = \mathbb{E}_{\mathbf{q},r,o}\left[\nabla_{\theta_i}\pi_{\theta_i}(\mathbf{q}_i)\nabla_{a_i}Q_w^\pi(\mathbf{q}_1, \cdots, \mathbf{q}_n, a_1, \cdots, a_n)|_{a_i=\pi_{\theta_i}(\mathbf{q}_i)}\right]$
$\mathcal{L}_2(\Theta_{\text{GAPSR}}, \Theta_{\text{MARL}}) = \sum_{i=1}^n E_{\tau \sim p(\tau|\Theta)}\left[\sum_j \left\| Z_{i,j}\left(F_{i,j}(W_{i,j}(\mathbf{q}_{t-1,i,j}))\otimes\phi_{t,j}^a\right) - \phi_{t,i}^o\right\|_2^2\right]$

33:     Get $W_{i,j}^k, Z_{i,j}^k$, and $\pi_i^k$

34: **end for**

35: **Output:** Return $\Theta = (\Theta_{GAPSR}, \Theta_{MARL})$

---

### B.4.1 GAPSR-1

It is based on the IAC (Foerster et al., 2018) which directly applies the single-agent policy gradient to have each agent learn independently, with the idea behind independent Q-learning (Tan, 1993), with actor-critic in place of Q-learning.

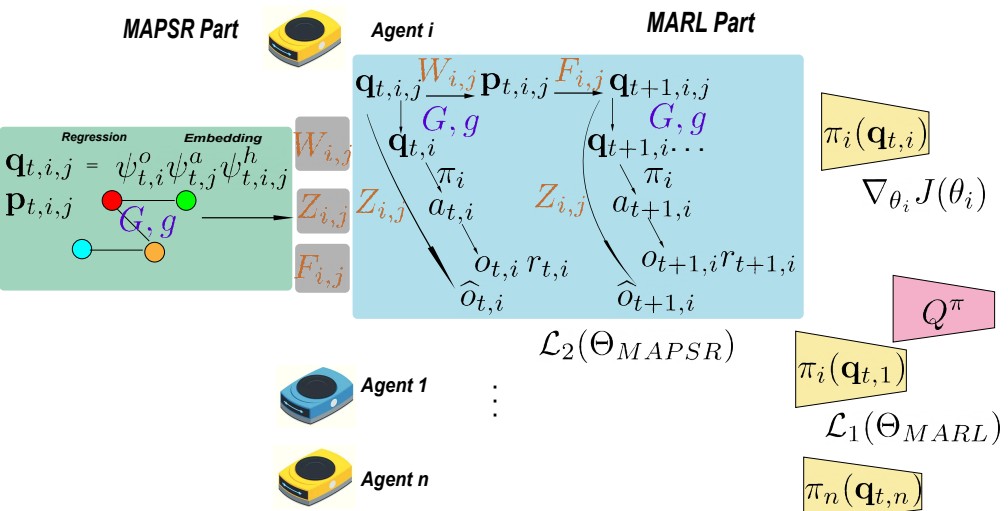

Figure 4: GAPSR architecture combining GAPSR and MARL. The left part is the GAPSR model, which corresponds to the GAPSR Model Estimation Phase in Algorithm 1. The right part is the MARL, which corresponds to the generation of predictive representations and learning the multi-agent actor-critic. We use the actor-critic framework, which contains a centralized critic and many decentralized actors. We could also give a parameter sharing actor network that maps individual PSR to parameters of a Gaussian distribution over the individual action space if agents are homogeneous. Both actor and critic are parameterized by neural networks. Section B.4 and section C.3 give a detailed description about this architecture.

**IAC** trains an actor-critic pair for each agent, resulting in actors $\pi_i(a_i|o_i)$ and critics $V_i(o_i, a_i)$.

$$\nabla_{\theta_i} J(\theta_i) = \mathbb{E}_{o,a,r} \Big[ \sum_{t=0}^{T} \nabla_{\theta_i} \log \pi_i(a_{t,i}|o_{t,i}) \big( r_i + \gamma V_i^{\pi_i}(o_{t+1,i}, a_{t+1,i}) - V_i^{\pi_i}(o_{t,i}, a_{t,i}) \big) \Big]. \quad (34)$$

While IAC agents display a strong ability to optimize individual rewards (Tan, 1993), the lack of global information and a mechanism for cooperation means they are likely to settle for sub-optimal solutions.

Here we use the predictive state $\mathbf{q}_i$ to fit the value and policy functions. And we train an actor-critic pair for each agent, resulting in actors $\pi_{\theta_i}(a_i|\mathbf{q}_i)$ and critics $V_i(\mathbf{q}_i)$.

$$\nabla_{\theta_i} J(\theta_i) = \mathbb{E}_{\tau_i \sim p(\tau_i|\Theta_i)} \Big[ \sum_{t=0}^{T} \nabla_{\theta_i} \log \pi_{\theta_i}(a_{t,i}|\mathbf{q}_{t,i})(r_i + \gamma V_{w_i}^{\pi_i}(\mathbf{q}_{t+1,i}) - V_{w_i}^{\pi_i}(\mathbf{q}_{t,i})) \Big]. \quad (35)$$

If the agents are homogeneous, we can share the critic and actor network. We have the critic loss as

$$\mathcal{L}_c = \mathbb{E}_{\tau_i \sim p(\tau_i|\Theta_i)}[(V_{w_i}^{\pi_i}(\mathbf{q}_i) - y_i)^2], \qquad y_i = r_i + \gamma \widehat{V}_{w_i}^{\pi_i}(\mathbf{q}_{t+1,i}). \quad (36)$$

Here $\widehat{V}$ is the target value function. Thus, the update of parameters is given by:

$$\delta_{t,i} = r_i + \gamma \widehat{V}^{\pi_i}(\mathbf{q}_{t+1,i}) - V^{\pi_i}(\mathbf{q}_i), \quad (37)$$

$$w_i' = w_i + \frac{1}{T} \sum_{t=0}^{T} \eta_w \delta_{t,i} \nabla_{w_i} V^{\pi_i}(\mathbf{q}_i), \quad (38)$$

$$\theta_i' = \theta_i + \frac{1}{T} \sum_{t=0}^{T} \eta_\theta \delta_{t,i} \nabla_{\theta_i} \log \pi_{\theta_i}(a_{t,i}|\mathbf{q}_{t,i}). \quad (39)$$

### B.4.2   GAPSR-2

The algorithm extends from MADDPG (Lowe et al., 2017) using the deterministic policy gradient of the PSR-value function.

MADDPG is an extension of deep deterministic actor-critic policy gradient (DDPG) (Lillicrap et al., 2015) to multi-agent setting such that let each agent's own critic is augmented with extra information about the actions of other agents, while their individual actor maintains a local state or observation.

The gradient of each agent is:
$$\nabla_{\theta_i} J(\theta_i) = \mathbb{E}_o[\nabla_{\theta_i} \pi_{\theta_i}(o_i) \nabla_{a_i} Q_i^\pi(o_1, \cdots, o_n, a_1, \cdots, a_n)|a_i = \pi(o_i)]. \tag{40}$$
The critic loss is
$$\mathcal{L} = \mathbb{E}_{o,a,r}[(Q_i^\pi(o_{t,1}, \cdots, o_{t,n}, a_{t,1}, \cdots, a_{t,n}) - y_i)^2] \quad y_i = r_i + \gamma \widehat{Q}_i^\pi(o_{t+1,1}, \cdots, o_{t+1,n}, a_{t+1,1}, \cdots, a_{t+1,n}), \tag{41}$$

where $\{\widehat{Q}_i^\pi\}$ is the set of target value functions with delayed parameters.

Our method considers the centralized critic. Also, in order to solve the non-stationary environments when each agent is learning, it uses the joint predictive states with joint actions as input.
$$\nabla_{\theta_i} J(\theta_i) = \mathbb{E}_{\mathbf{q},r,o}[\nabla_{\theta_i} \pi_{\theta_i}(\mathbf{q}_i) \nabla_{a_i} Q_w^\pi(\mathbf{q}_1, \cdots, \mathbf{q}_n, a_1, \cdots, a_n)|a_i = \pi(\mathbf{q}_i)] \tag{42}$$

The critic loss is defined as below:
$$\begin{aligned} \mathcal{L}_c = \quad & \mathbb{E}_{\mathbf{q},r,o}[(Q_w^\pi(\mathbf{q}_{t,1}, \cdots, \mathbf{q}_{t,n}, a_{t,1}, \cdots, a_{t,n}) - y)^2], \\ y = \quad & r + \gamma \widehat{Q}_w^\pi(\mathbf{q}_{t+1,1}, \cdots, \mathbf{q}_{t+1,n}, a_{t+1,1}, \cdots, a_{t+1,n}). \end{aligned} \tag{43}$$

Thus, the update of parameters is given by:

$$\delta_t = r_t + \gamma \widehat{Q}^\pi(\mathbf{q}_{t+1,1}, \cdots, \mathbf{q}_{t+1,n}, a_{t+1,1}, \cdots, a_{t+1,n}) - Q^\pi(\mathbf{q}_{t,1}, \cdots, \mathbf{q}_{t,n}, a_{t,1}, \cdots, a_{t,n}), \tag{44}$$

$$w' = w + \frac{1}{T} \sum_{t=0}^T \eta_w \delta_t \nabla_w Q^\pi(\mathbf{q}_{t,1}, \cdots, \mathbf{q}_{t,n}, a_{t,1}, \cdots, a_{t,n}), \tag{45}$$

$$\theta_i' = \theta_i + \frac{1}{T} \sum_{t=0}^T \eta_\theta \nabla_{\theta_i} \pi_{\theta_i}(a_i|\mathbf{q}_i) \nabla_{a_i} Q^\pi(\mathbf{q}_1, \cdots, \mathbf{q}_n, a_1, \cdots, a_n)|_{a_i = \pi(\mathbf{q}_i)}. \tag{46}$$

## C  ENVIRONMENT AND EXPERIMENT

### C.1  MAMUJOCO ENVIRONMENT SETUP

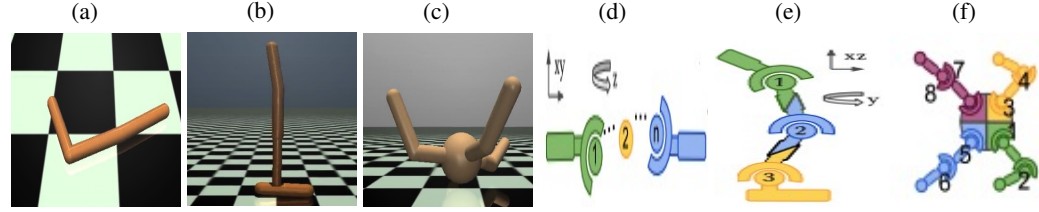

Figure 5: The illustration of three environments swimmer, hopper, ant, and their corresponding MAMuJoCo version. (a) Single swimmer; (b) Single hopper; (c) Single ant; (d) n-agents swimmer; (e) 3-agents hopper; (f) 4-agents ant.

Here, we give the details of setting up our multi-agent environment - MAMuJoCo. As introduced in section 6, the many agents are constructed by separating a existed single agent into parts, and each agent will only control a part of the whole agent (Figure 5).

**Partially Observable space**: MAMuJoCo is a simulated robotic environment, the partially observable property is achieved by only allowing partial information to the agents. For all environments, only the angles of the agent's joints are visible to the network; the velocities are hidden.

**Action**: Each agent's action space in MAMuJoCo is given by the joint action space overall motors controllable by that agent.

**Observation**: For each agent $i$, observations are constructed by inferring which body segments and joints are observable by an agent $i$. Each agent can always observe all joints within its sub-graph. A configurable parameter $k \geq 0$ determines the maximum graph distance to the agent's subgraph at which joints are observable. For example, $k = 0$ means agents can only observe their own joints and body parts, while $k = 1$ means it can observe its adjacent joints, which has 1 graph distance to the

Table 1: Configurations for MAMuJoCo environment

| | Environments | | |
|---|---|---|---|
| | **Swimmer** | | |
| $k$ | $0$ | | |
| $R$ | $\sum_i(\frac{\Delta d_i}{\Delta t}) + 0.0001r$ | | |
| $r$ | $r = -\|\mathbf{a}\|_2^2$ is a regularizer for joint action $\mathbf{a}$ | | |
| | **Hopper** | | |
| $k$ | $2$ | | |
| $R$ | $\sum_i(\frac{\Delta d_i}{\Delta t}) + 0.001r + 1.0$ | | |
| $r$ | $r = -\|\mathbf{a}\|_2^2$ is a regularizer for joint action $\mathbf{a}$ | | |
| | **Ant** | | |
| $k$ | $0$ | | |
| $R$ | $\sum_i(\frac{\Delta d_i}{\Delta t}) + 5 \cdot 1e(-3)\,\|\text{external contact forces}\|_2^2 + 0.0001r$ | | |
| $r$ | $r = -\|\mathbf{a}\|_2^2$ is a regularizer for joint action $\mathbf{a}$ | | |

agent. The agent observation is then given by a fixed order concatenation of each observable graph element's representation vector. Depending on the environment and configuration, representation vectors may include attributes such as position, velocity, and external body forces. In addition to joint and body segment-specific observation categories, agents can also be configured to observe the robot's central torso's position and velocity attributes.

## C.2 Experiment setup

We select three experiments from MAMuJoCo and give a detailed description of the experiments' setup. In all environments, the agent has the goal to maximize the velocity of the first coordinate for the team. We use $k$ to denote the maximum observation distances to the subgraph. We use $\Delta d$ to denote the first coordinate position difference between a time difference $\Delta t$. Finally, we use $R$ to denote the reward function. Table 1 has the configuration details for these parameters.

## C.3 Neural Network Architecture

We implement all algorithms using deep neural networks as function approximators. We ensure that all policy and and action-value functions have the same neural network architecture among all algorithms to the extent each algorithm allows for a fair comparison.

Usually, in a continuous environment, each agent's policy will be parameterized by its actor network that outputs the mean and diagonal covariance of a Gaussian distribution over the continuous action space. For our experiments with continuous action spaces, a Gaussian distribution with a diagonal covariance matrix is used. The policy network maps from the input feature to a Gaussian distribution vector $\mu$. Moreover, $\mu = [\text{mean, std}]$, where mean is a vector specifies the action means, and std vector specifies the standard deviation. For deterministic policy, it maps to the action vector.

In the implementation of actor-critic method, all the actor-network is parameterized by a multi-layer perceptron (MLP) with two hidden layers of size 400 and 300 respectively and ReLU activation, which takes in the individual agent's predictive state and outputs the mean and covariance of a Gaussian policy for stochastic policy, or the action vector for deterministic policy. The critic network is also an MLP with two hidden layers with 400 and 300 units, respectively. For GAPSR-1, the critic network is used to approximate per-agent utilities, which receives each agent's predictive state as input. For IAC, same as GAPSR-1, it receives agent local observation and individual action as input.

In GAPSR-2, there is a shared critic network that approximates all agents utilities, which receives all agents' predictive states and the joint action of all agents as input. The global state consists of the complete state information from the environment. In MADDPG, the critic receives the global state and the joint action of all agents as input. Each decentralized actor (i.e., policy) network takes in each agent's observation and outputs the agent's action vector.

Table 2: Model parameters for MAMuJoCo environment

|  | Environments | | |
|---|---|---|---|
|  | Swimmer | Hopper | Ant |
| $n$ | 2 | 3 | 4 |
| $\mu$ | 0 | 0 | 0 |
| $\sigma$ | 0.1 | 0.1 | 0.1 |
| $\gamma$ | 0.99 | 0.99 | 0.99 |
| Soft target network | 0.001 | 0.001 | 0.001 |
| $\alpha_1$ | 0.7 | 0.65 | 0.7 |
| $\alpha_2$ | 0.3 | 0.35 | 0.3 |
| $\beta$ | 0.6 | 0.45 | 0.5 |
| $\eta$ learning rate of Adam | 0.001 | 0.001 | 0.001 |
| $\lambda$ ridge regression regularization | 0.01 | 0.01 | 0.01 |
| Total iterations | 50 | 50 | 50 |
| Number of trajectories | 100 | 100 | 100 |
| Maximum number of steps per trajectory | 1000 | 1000 | 1000 |
| Length of test window | 8 | 12 | 10 |
| Length of history window | 8 | 12 | 10 |

## C.4 MODEL PARAMETERS

The hyper parameters for the the MAMuJoCo environments are in Table 2.

## D SUPPLEMENT EXPERIMENTS

### D.1 MAMUJOCO SUPPLEMENT EXPERIMENTAL RESULTS

We plotted the predictive observations compared to actual observations in Figure 2 at the beginning of the learning process (iteration 1) and end of the learning (iteration 40). We also show the results of iteration ten and iteration 20 in Figure 6. By comparing with iterations 1 and iterations 40 in Figure 2, we see that the iteration 40 has the smallest difference between predictive observation and true observation, and the difference gets increased as the iteration goes to the earlier stage of the learning process. So the predictive accuracy is improved incrementally with the learning progresses.

### D.2 MULTI-AGENT PARTICLE ENVIRONMENT

We also test GAPSR into another environment, multi-agent particle environment (Lowe et al., 2017). The agents are displaced into a 2-dimensional coordinate. This environment does not assume that all agents have identical action and observation spaces. We run experiments using a different number of agents on two environments, the predator-prey, and cooperative-push. We use the same configuration in (Liu et al., 2020a). The network has the same design as MAMuJoCo in section C.3, except we use MLP with two hidden layers with the same 128 units respectively, so we do not repeat the description. In Table 3, we report the hyperparameters for multi-agent particle environments.

In the supplementary materials we provide the code and instructions.

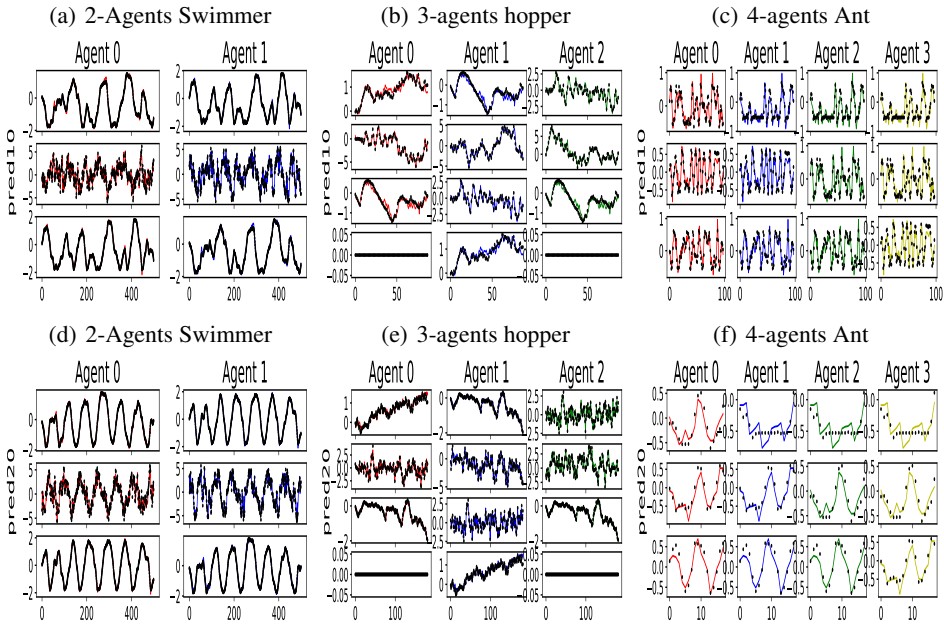

Figure 6: Additional Experiments Results, Predicted Trajectories vs Actual Observations for Multi-Agent Environments. (a) - (c) Iteration 10; (d) - (f) Iteration 20; This is the supplement for Figure 2

Table 3: Model parameters for multi-agent particle environment

| | Environments | | | | | |
| | Predator-prey | | | Cooperative-push | | |
| | n=3 | n=15 | n=100 | n=3 | n=15 | n=30 |
|---|---|---|---|---|---|---|
| $\gamma$ | 0.99 | 0.99 | 0.99 | 0.99 | 0.99 | 0.99 |
| Soft target network | 0.001 | 0.001 | 0.001 | 0.001 | 0.001 | 0.001 |
| $\alpha_1$ | 0.8 | 0.75 | 0.5 | 0.8 | 0.75 | 0.7 |
| $\alpha_2$ | 0.2 | 0.25 | 0.5 | 0.2 | 0.25 | 0.3 |
| $\beta$ | 0.45 | 0.5 | 0.45 | 0.55 | 0.5 | 0.5 |
| $\eta$ learning rate of Adam | 0.05 | 0.05 | 0.05 | 0.05 | 0.05 | 0.05 |
| $\lambda$ ridge regression regularization | 0.01 | 0.01 | 0.01 | 0.01 | 0.01 | 0.01 |
| Total iterations | 30 | 30 | 30 | 30 | 30 | 30 |
| Number of trajectories | 100 | 100 | 100 | 100 | 100 | 100 |
| Maximum number of steps per trajectory | 500 | 500 | 500 | 500 | 500 | 500 |
| Length of test window | 5 | 5 | 5 | 5 | 5 | 5 |
| Length of history window | 5 | 5 | 5 | 5 | 5 | 5 |

## E DETAILED PROOFS

We give the proof of Theorem 1 in Appendix E.1, and we first introduce the following Lemmas to prepare the proof.

**Definition 2.** *Let $X_1 \ldots X_k$ be independent random variables of dimensionality $d_{X_1} \ldots d_{X_k}$ such that $\|X_k\| < t_{x_k}$. Let $\{(x_{1_j} \ldots, x_{k_j})\}_{j=1}^N$ be the $N$ i.i.d samples from distribution of $X_1, \ldots, X_k$, the $C_{X_1} := \mathbb{E}[X_1 X_1^T]$ and $\widehat{C}_{X_1} = \frac{1}{N} \sum_{j=1}^N x_{1_j} x_{1_j}^T$, and $C_{X_1 X_2} := \mathbb{E}[X_1 X_2^T]$ and $\widehat{C}_{X_1, X_2} = \frac{1}{N} \sum_{j=1}^N x_{1_j} x_{2_j}^T$. Also, we use the $(\cdot), v(\cdot)$ to denote the largest, smallest eigenvalue of a matrix.*

**Lemma 3** ((Tropp, 2015)). *Let $a_j$ be a finite sequence of independent random, Hermitian matrices with dimension $d$. Assume that $0 \leq v(a_j)$ and $u(a_j) \leq L$ for each $j$. Let $S = \sum_j a_j$, then for any $\eta \in [0, 1]$, it follows that*

$$\mathbf{Pr}(v(S) \leq (1-\eta)v(\mathbb{E}[S])) \leq d\left[\frac{e^{-\eta}}{(1-\eta)^{1-\eta}}\right]^{v(\mathbb{E}[S])/L} \leq 2de^{-\eta v(\mathbb{E}[S])/L}. \qquad (47)$$

**Corollary 1.** *Let $X$ be a random variable, for any $\epsilon \in (0,1)$ such that $N > \frac{t_x^2 \log(2d_X/\epsilon)}{v(C_X)}$ the following holds with probability at least $1 - \epsilon$*

$$v(\widehat{C}_X) > \frac{t_x^2 \log(2d_X/\epsilon)}{v(C_X)N}.$$

*In other words, if $N$ large enough, then $\widehat{C}_X$ and $C_X$ will be close enough.*

*Proof.* Define $S_j = 1/N x_j x_j^T$. Then it follows that $u(S_j) \leq L = t_x^2/N$ and define $\epsilon := 2d_X e^{-\sigma N v(C_X)/t_x^2}$, which implies that $\sigma = \frac{t_x^2 \log(2d_X/\epsilon)}{v(C_X)N}$. Then it follows from Matrix Chernoff Inequality in Lemma 3 that $P_r(v(\widehat{C}_X) \leq (1-\sigma)v(C_X)) \leq \epsilon$. $\square$

**Lemma 4** ((Tropp, 2015))**.** *A finite sequence $\{a_j\}$ of independent, random matrices with common dimensions $a \times b$, and assume that $\mathbb{E}[a_j] = 0$ and $\|a_j\| \leq L$ for each $j$, let $S = \sum_j a_j$ as a random matrix. Let $Var(S)$ be the variance statistics such that $Var(S) = \max\{\|\mathbb{E}[SS^T]\|, \|\mathbb{E}[S^TS]\|\}$, then*

$$\mathbf{Pr}(\|S\| > c) \leq (a+b)e^{\frac{-c^2/2}{Var(S)+Lc/3}}. \tag{48}$$

**Corollary 2.** *With at least probability $1 - \epsilon$ that*

$$\left\|\widehat{C}_{YX} - C_{YX}\right\| \leq \sqrt{\frac{2\log(d_Y + d_X)/\epsilon Var}{N}} + \frac{2\log((d_Y + d_X)/\epsilon)L}{3N}.$$

*where $L = t_y t_x + \|C_{YX}\| \leq 2t_y t_x$ and $Var = \max\{t_y^2 \|C_X\|, t_x^2 \|C_Y\|\} + \|C_{YX}\|^2 \leq 2t_y^2 t_x^2$.*

*Proof.* Let $X, Y$ be two random variables, and let a finite sequence $\{a_j\}$ of independent random matrices to satisfy $a_j = y_j x_j^T - C_{YX}$. So the $a_j$ will have dimensions $d_X \times d_Y$. Let random matrix $S = \sum_j a_j$. It follows that $\mathbb{E}[a_j] = 0$ and $\|a_j\| = \|y_j x_j^T - C_{YX}\| \leq \|y_k\| \|x_k\| + \|C_{YX}\| \leq t_y t_x + \|C_{YX}\|$,

$$\left\|\mathbb{E}[SS^T]\right\| = \left\|\sum_{i,j}\left(\mathbb{E}[y_i x_i^T x_j y_j^T] - C_{YX}C_{XY}\right)\right\|$$

$$= \left\|\sum_i\left(\mathbb{E}[\|x_i\|^2 y_i y_i^T] - C_{YX}C_{XY}\right) + \sum_{i \neq j}\left(\mathbb{E}[y_i x_i^T]\mathbb{E}[x_j y_j^T] - C_{YX}C_{XY}\right)\right\|$$

$$= \left\|\sum_i(\mathbb{E}[\|x_i\|^2 y_i y_i^T] - C_{YX}C_{XY})\right\|$$

$$\leq N(t_x^2 \|C_Y\| + \|C_{YX}\|^2).$$

Similarly, $\|\mathbb{E}[SS^T]\| \leq N(t_y^2 \|C_X\| + \|C_{YX}\|^2)$. By applying lemma 4, we have $\epsilon = \mathbf{Pr}(\|S\| \geq Nc) \leq (d_X + d_Y)e^{\left(\frac{-Nc^2/2}{Var+Lc/3}\right)}$ and therefore, it implies that

$$c \leq \frac{\log((d_X + d_Y)/\epsilon)L}{3N} + \sqrt{\frac{(\log(d_X + d_Y/\epsilon))^2 L^2}{9N^2} + \frac{2\log((d_X + d_Y)/\epsilon)Var}{N}}$$

$$\leq \frac{2\log((d_X + d_Y)/\epsilon)L}{3N} + \sqrt{\frac{2\log((d_X + d_Y)/\epsilon)Var}{N}}.$$

$\square$

**Corollary 3.** *For random variable $X$ with dimensionality $d_X$ and $\|X\| \leq t_x$, with probability $1 - \epsilon$, it follows that*

$$\left\|C_X^{-1/2}(\widehat{C}_X) - C_X\right\| \leq 2t_x\sqrt{\frac{2\log(2d_X/\epsilon)}{N}} + \frac{2\log(2d_X/\epsilon)L}{3N},$$

*where $L = \frac{t_x^2}{\sqrt{v(C_X)}} + t_x$*

*Proof.* The proof is similarly to the the proof of corollary 2, define $a_j = \sum_X^{-1/2} x_j x_j^T - C_X^{1/2}$, $S = \sum a_j$ then it follows that $\mathbb{E}[a_j] = 0$ and $\|a_j\| \leq \frac{t_x^2}{\sqrt{v(C_X)}} + t_x$,

$$\left\|\mathbb{E}[S^T S]\right\| = \left\|\mathbb{E}[SS^T]\right\| \leq N(t_x^2 + \|C_X\|^2) \leq 2Nt_x^2. \tag{49}$$

Applying lemma 4 to get

$$\epsilon = \mathbf{Pr}(\|S\| \geq Nc) \leq 2d_X e^{\frac{-Nc^2/2}{2t_x^2 + Lc/3}}, \tag{50}$$

it follows that

$$c \leq \frac{2\log(2d_X/\epsilon)L}{3N} + 2t_x \sqrt{\frac{\log(2d_X/\epsilon)}{N}}. \tag{51}$$

$\square$

**Lemma 5.** *For two random variables $X, Y$, let $\widehat{C}_{YX} = C_{YX} + \Delta_{YX}$, and $\widehat{C}_X = C_X + \Delta_X$ where $\mathbb{E}[\Delta_{YX}]$ and $\mathbb{E}[\Delta_X]$ are not necessarily zero and $\widehat{C}_X$ is symmetric positive semidefinite. Define $A = C_{YX} C_X^{-1}$ and $\widehat{A} = \widehat{C}_{YX}(\widehat{C}_X + \lambda)^{-1}$. Then it follows that:*

$$\left\|\widehat{A} - A\right\| \leq \sqrt{\frac{u(C_Y)}{v(C_X)}} \left( \frac{\sqrt{v(C_X)}\left\|C_X^{-1/2}\right\|\Delta_X + \lambda}{v(\widehat{C}_X) + \lambda} \right) + \frac{\|\Delta_{YX}\|}{v(\widehat{C}_X) + \lambda}.$$

*Proof.*

$$\widehat{A} - A = C_{YX}((C_X + \Delta_X + \lambda I)^{-1} - C_X^{-1}) + \Delta_{YX}(C_X + \Delta_X + \lambda I)^{-1} = M_1 + M_2.$$

It follows that

$$\|M_2\| \leq \frac{\Delta_{YX}}{v(\widehat{C}_X) + \lambda}.$$

For $M_1$, by using facts $U^{-1} - V^{-1} = U^{-1}(V - U)V^{-1}$ and $C_{YX} = C_Y^{1/2} P C_X^{1/2}$, where $P$ is a correlation matrix with $\|P\| \leq 1$,

$$M_1 = -C_{YX} C_X^{-1}(\Delta_X + \lambda I)(C_X + \Delta_X + \lambda I)^{-1}$$
$$= -C_Y^{1/2} P C_X^{-1/2}(\Delta_X + \lambda I)(C_X + \Delta_X + \lambda I)^{-1},$$

$$\|M_1\| \leq \sqrt{u(C_Y)} \frac{\left\|C_X^{-1/2}\Delta_X\right\| + \lambda\left\|C_X^{-1/2}\right\|}{v(\widehat{C}_X) + \lambda}$$
$$\leq \sqrt{\frac{u(C_Y)}{v(C_X)}} \frac{\sqrt{v(C_X)}\left\|C_X^{-1/2}\Delta_X\right\| + \lambda}{v(\widehat{C}_X) + \lambda}.$$

$\square$

**Corollary 4.** *Let $\{(x_k, y_k)\}_{k=1}^N$ be i.i.d samples from two random variables $X, Y$ with dimensions $d_X$ and $d_Y$ and (uncentered) covariances $C_X$ and $C_Y$. Assume $\|X\| \leq t_x$ and $\|Y\| \leq t_y$. Define $A = C_{YX} C_X^{-1}$ and $\widehat{A} = \widehat{C}_{YX}(\widehat{C}_X + \lambda)^{-1}$. For any $\epsilon \in (0, 1)$ such that $N > \frac{t_x^2 \log(2d_X/\epsilon)}{v(C_X)}$ the following holds with probability at least $1 - 3\epsilon$:*

$$\left\|\widehat{A} - A\right\| \leq \sqrt{\frac{u(C_Y)}{v(C_X)}} \left( \frac{\sqrt{v(C_X)}\alpha + \lambda}{v(C_X)(1 - \gamma) + \lambda} \right) + \frac{\beta}{v(C_X)(1 - \gamma) + \lambda},$$

*where*

$$\alpha = 2t_x \sqrt{\frac{2\log(2d_X/\epsilon)}{N}} + \frac{2\log(2d_X/\epsilon)}{3N}\left(\frac{c_x^2}{\sqrt{v(C_X)}} + t_x\right),$$

$$\beta = 2t_y t_x \sqrt{\frac{\log(d_Y + d_X)/\epsilon}{N}} + \frac{3t_y t_x \log((d_Y + d_X)/\epsilon)}{3N},$$

$$\gamma = \frac{t_x^2 \log(2d_X/\epsilon)}{v(C_X)N}.$$

*Proof.* It follows by applying Corollaries 1,2,3 to Lemma 5. By union bound, each condition has probability $1 - \epsilon$, so the total events are bounded by

$$\mathbf{Pr}(\text{bounds satisfied}) := 1 - \mathbf{Pr}(\bigcup_{i=1}^{3} A_i) \geq 1 - \sum_{i=1}^{3} \mathbf{Pr}(A_i) = 1 - 3\epsilon.$$

$\square$

**Lemma 6.** *For two random variables $X, Y$, let $\widehat{C}_{YX} = C_{YX} + \Delta_{YX}$, and $\widehat{C}_X = C_X + \Delta_X$ where $\mathbb{E}[\Delta_{YX}]$ and $\mathbb{E}[\Delta_X]$ are not necessarily zero and $\widehat{C}_X$ is symmetric but not positive semidefinite. Define $A = C_{YX} C_X^{-1}$ and $\widehat{A} = \widehat{C}_{YX} \widehat{C}_X (\widehat{C}_X^2 + \lambda I)^{-1}$. Then it follows that:*

$$\left\| \widehat{A} - A \right\| \leq \sqrt{\frac{u(C_Y)}{v(C_X)^3}} \frac{\|\Delta_x\|^2 + 2u(C_X)\|\Delta_X\| + \lambda}{v(\widehat{C}_X) + \lambda} + \frac{\|C_{YX}\|\|\Delta_X\| + \|\Delta_{YX}\|\|C_X\| + \|\Delta_{YX}\|\|\Delta_X\|}{v(\widehat{C}_X)^2 + \lambda}.$$

*Proof.*
$$\widehat{A} - A = (C_{YX} + \Delta_{yx})(C_X + \Delta_X)((C_X + \Delta_X)^2 + \lambda I)^{-1} - C_{YX} C_X C_X^{-2}$$
$$= C_{YX} C_X (((C_X + \Delta_X)^2 + \lambda I)^{-1} - C_X^{-2}) + (C_{YX}\Delta_X + \Delta_{YX}C_X + \Delta_{YX}\Delta_X)((C_X + \Delta_X)^2 + \lambda I)$$
$$= M_1 + M_2.$$

For $M_1$, by using facts $U^{-1} - V^{-1} = U^{-1}(V - U)V^{-1}$ and $C_{YX} = C_Y^{1/2} P C_X^{1/2}$, where $P$ is a correlation matrix with $\|P\| \leq 1$, it follows that
$$M_1 = -C_Y^{1/2} P C_X^{-3/2}(\Delta_X^2 + C_X\Delta_X + \Delta_X C_X + \lambda I)((C_X + \Delta_X)^2 + \lambda I)^{-1}.$$
Therefore,

$$\|M_1\| \leq \sqrt{\frac{u(C_Y)}{v(C_X)^3}} \frac{\|\Delta_x\|^2 + 2u(C_X)\|\Delta_X\| + \lambda}{v(\widehat{C}_X) + \lambda},$$

$$\|M_2\| \leq \frac{\|C_{YX}\|\|\Delta_X\| + \|C_{YX}\|\|C_X\| + \|C_{YX}\|\|\Delta_X\|}{v(\widehat{C}_X)^2 + \lambda}.$$

$\square$

**Corollary 5.** *Let $\{(x_k, y_k)\}_{k=1}^{N}$ be i.i.d samples from two random variables $X, Y$ with dimensions $d_X$ and $d_Y$ and (uncentered) covariances $C_X$ and $C_Y$. The $\mathbb{E}[\Delta_{YX}]$ and $\mathbb{E}[\Delta_X]$ is not necessarily zero and $C_x$ is symmetric but not necessarily positive semidefinite. Assume $\|X\| \leq t_x$ and $\|Y\| \leq t_y$. Define $A = C_{YX} C_X^{-1}$ and $\widehat{A} = \widehat{C}_{YX} \widehat{C}_X (\widehat{C}_X^2 + \lambda)^{-1}$. For any $\epsilon \in (0, 1)$ such that $N > \frac{t_x^2 \log(2d_X/\epsilon)}{v(C_X)}$ the following holds with probability at least $1 - 3\epsilon$:*

$$\left\| \widehat{A} - A \right\| \leq \sqrt{\frac{u(C_Y)}{v(C_X)^3}} \frac{\|\Delta_x\|^2 + 2u(C_X)\|\Delta_X\| + \lambda}{v(C_X)(1 - \gamma) + \lambda} + \frac{\|C_{YX}\|\|\Delta_X\| + \|\Delta_{YX}\|\|C_X\| + \|\Delta_{YX}\|\|\Delta_X\|}{v(C_X)^2(1 - \gamma)^2 + \lambda},$$

*where*

$$\gamma = \frac{t_x^2 \log(2d_X/\epsilon)}{v(C_X)N}.$$

*Proof.* It follows by applying Corollaries 1,2,3 to Lemma 6. Also by union bound, so the total events are bounded $1 - 3\epsilon$.

$\square$

**Theorem 2.** *Grabner 1997 (3.3) (Grabner & Prodinger, 1997) Consider there are $n$ independent copies $X_1, \ldots X_n$ i.i.d negative binomial random variables, with parameters defined as $\mathcal{NB}(b, p)$, and our goal is to calculate the expectation of the maximum of these $N$ random variables $\mathbb{E}_n = \mathbb{E}\{\max(X_1, \ldots, X_n)\}$ then we have following asymptotic solution:*

$$\mathbb{E}_n = \log_{\frac{1}{q}}(n) + (b - 1)\log_{\frac{1}{q}}\log_{\frac{1}{q}}(n) + (b - 1)\log_{\frac{1}{q}} p + (b - 1) - \log_{\frac{1}{q}}(b - 1)! + \frac{1}{2} + \frac{\gamma}{\log_{\frac{1}{q}}(1/q)}$$

$$+ F(\log_{\frac{1}{q}}(n) + (b - 1)\log_{\frac{1}{q}}\log_{\frac{1}{q}}(n) + (b - 1)\log_{\frac{1}{q}} p - \log_{\frac{1}{q}}(b - 1)!) + o(1), \tag{52}$$

*(where $F$ is a periodic $C^\infty - function$ of period 1 and mean value 0 whose Fourier-coefficients are given by $\hat{F}(k) = -\frac{1}{\log(\frac{1}{q})}\Gamma(-\frac{2k\pi i}{\log(\frac{1}{q})})$ for $k \in \mathbb{Z} \setminus \{0\}$, and $q = 1 - p$).*

We omit the proof, interested readers could go to (Grabner & Prodinger, 1997) for details.

### E.1 PROOF OF THEOREM 1

We first prove the bound for $\|\widehat{\mathbf{q}}_{i,j} - \mathbf{q}_{i,j}\|$.

**Proposition 2.** *Let $\pi_\Theta$ be a data collection policy and $\mathcal{H}$ is the range of $\pi_\Theta$ on joint histories. If Equation 7 used, then for all $h \in \mathcal{H}$ and any $\epsilon \in (0,1)$, $\left\|\widehat{\mathbf{q}}_{i,j}(\psi^h) - \mathbf{q}_{i,j}(\psi^h)\right\|$ is bounded as below with probability at least $1 - 3\epsilon$.*

$$\|\widehat{\mathbf{q}}_{i,j} - \mathbf{q}_{i,j}\| \leq \sqrt{\frac{u\left(C_{\psi_i^o|\psi_{i,j}^h}\right)}{v\left(C_{\psi_j^a|\psi_{i,j}^h}\right)^3}\frac{\|\Delta_1\|^2 + 2u\left(C_{\psi_j^a|\psi_{i,j}^h}\right)\|\Delta_1\| + \lambda}{v\left(C_{\psi_j^a|\psi_{i,j}^h}\right)(1-\gamma) + \lambda}}$$

$$+ \frac{\left\|C_{\psi_i^o\psi_j^a|\psi_{i,j}^h}\right\|\|\Delta_1\| + \|\Delta_2\|\left\|C_{\psi_j^a|\psi_{i,j}^h}\right\| + \|\Delta_2\|\|\Delta_1\|}{v\left(C_{\psi_j^a|\psi_{i,j}^h}\right)^2(1-\gamma)^2 + \lambda},$$

*where $\Delta_1$ follows the bound (55) and $\Delta_2$ follows the bound (53), and $\gamma = \frac{t_{A_j}^2\log(2d_{A_j}/\epsilon)}{v\left(C_{\psi_j^a}\right)N}$.*

*Proof.* Let $T_{i,j}$ is the tensor such that $C_{\psi_i^o\psi_j^a|\psi_{i,j}^h} = T_{i,j} \times_h \psi_{i,j}^h$, and $U_{i,j}$ is the tensor such that $C_{\psi_j^a|\psi_{i,j}^h} = U_{i,j} \times_h \psi_{i,j}^h$ and for simplicity without loss meaning, we use $C_{\psi_j^a|\psi_{i,j}^h}$ to denote $C_{\psi_j^a\psi_j^a|\psi_{i,j}^h}$. Then we have

$$\left\|\widehat{C}_{\psi_i^o\psi_j^a|\psi_{i,j}^h} - C_{\psi_i^o\psi_j^a|\psi_{i,j}^h}\right\| \leq \left\|\widehat{T}_{i,j} - T_{i,j}\right\|\left\|\psi_{i,j}^h\right\|,$$

$$\left\|\widehat{C}_{\psi_j^a|\psi_{i,j}^h} - C_{\psi_j^a|\psi_{i,j}^h}\right\| \leq \left\|\widehat{U}_{i,j} - U_{i,j}\right\|\left\|\psi_{i,j}^h\right\|.$$

We finish the above proof by proofing the $\|T_{i,j} - T_{i,j}\|$ and $\|U_{i,j} - U_{i,j}\|$ are bounded by using Corollary 4.

$$\left\|\widehat{T}_{i,j} - T_{i,j}\right\|\left\|\psi_{i,j}^h\right\| \leq t_h\sqrt{\frac{u\left(C_{\psi_i^o,\psi_j^a}\right)}{v\left(C_{\psi_{i,j}^h}\right)}}\left(\frac{\sqrt{v\left(C_{\psi_{i,j}^h}\right)}\alpha + \lambda}{v\left(C_{\psi_{i,j}^h}\right)(1-\gamma) + \lambda}\right) + \frac{\beta}{v\left(C_{\psi_{i,j}^h}\right)(1-\gamma) + \lambda},$$

$$\tag{53}$$

where

$$\alpha = 2t_h\sqrt{\frac{2\log(2d_h/\epsilon)}{N}} + \frac{2\log(2d_h/\epsilon)}{3N}\left(\frac{t_h^2}{\sqrt{v\left(C_{\psi_{i,j}^h}\right)}} + t_h\right),$$

$$\beta = 2t_{O_i}t_{A_j}t_h\sqrt{\frac{\log(d_{O_i}d_{A_j} + d_h)/\epsilon}{N}} + \frac{4t_{O_i}t_{A_j}t_h\log((d_{O_i}d_{A_j} + d_h)/\epsilon)}{3N},$$

$$\gamma = \frac{t_h^2\log(2d_h/\epsilon)}{v\left(C_{\psi_{i,j}^h}\right)N};$$

$$\tag{54}$$

and

$$\left\|\widehat{U}_{i,j} - U_{i,j}\right\| \left\|\psi_{i,j}^h\right\| \le t_h \sqrt{\frac{u\left(C_{\psi_j^a}\right)}{v\left(C_{\psi_{i,j}^h}\right)}} \left(\frac{\sqrt{v\left(C_{\psi_{i,j}^h}\right)}\alpha + \lambda}{v\left(C_{\psi_{i,j}^h}\right)(1-\gamma)+\lambda}\right) + \frac{\beta}{v\left(C_{\psi_{i,j}^h}\right)(1-\gamma)+\lambda},$$

$$(55)$$

where

$$\alpha = 2t_h \sqrt{\frac{2\log(2d_h/\epsilon)}{N}} + \frac{2\log(2d_h/\epsilon)}{3N}\left(\frac{t_h^2}{\sqrt{v\left(C_{\psi_{i,j}^h}\right)}} + t_h\right),$$

$$\beta = 2t_{A_j}t_h\sqrt{\frac{\log(d_{A_j}+d_h)/\epsilon}{N}} + \frac{4t_{A_j}t_h\log((d_{A_j}+d_h)/\epsilon)}{3N},$$

$$\gamma = \frac{t_h^2\log(2d_h/\epsilon)}{v\left(C_{\psi_{i,j}^h}\right)N}.$$

$$(56)$$

Then using the equation (7) and corollary 5 to obtain the bound for $\mathbf{q}_{i,j}$

$$\|\widehat{\mathbf{q}}_{i,j} - \mathbf{q}_{i,j}\| \le \sqrt{\frac{u\left(C_{\psi_j^o|\psi_{i,j}^h}\right)}{v\left(C_{\psi_j^a|\psi_{i,j}^h}\right)^3}} \frac{\|\Delta_1\|^2 + 2u\left(C_{\psi_j^a|\psi_{i,j}^h}\right)\|\Delta_1\| + \lambda}{v\left(C_{\psi_j^a|\psi_{i,j}^h}\right)(1-\gamma)+\lambda}$$

$$+ \frac{\left\|C_{\psi_i^o\psi_j^a|\psi_{i,j}^h}\right\|\|\Delta_1\| + \|\Delta_2\|\left\|C_{\psi_j^a|\psi_{i,j}^h}\right\| + \|\Delta_2\|\|\Delta_1\|}{v\left(C_{\psi_j^a|\psi_{i,j}^h}\right)^2(1-\gamma)^2+\lambda},$$

where $\Delta_1$ follows the bound (55) and $\Delta_2$ follows the bound (53), and $\gamma = \frac{t_{A_j}^2\log(2d_{A_j}/\epsilon)}{v\left(C_{\psi_j^a}\right)N}$.

$\square$

Now we start to prove our Theorem 1.

*Proof.* The Equation (5) says:

$$\mathbf{q}_{t,i} \coloneqq g\left(\{\mathbf{q}_{t,i,j}\}_{j=1}^n\right) = \sum_j \mathbf{q}_{t,i,j},$$

here we assume the agents are homogeneous, in other words, each pair of $\{(o_i, a_i)\}_{i=1}^n$ coming from the same spaces $\mathcal{O}, \mathcal{A}$. They are permutation invariant and their identities do not matter. Thus, the bound of $\mathbf{q}_{i,j}$ is invariant to agents. Under the assumption of static fully complete graph, for any agent $\mathbf{q}_i = \sum_{j=1}^n \mathbf{q}_{i,j}$, thus

$$\begin{aligned}\|\widehat{\mathbf{q}}_i - \mathbf{q}_i\| &= \left\|\sum_j \widehat{\mathbf{q}}_{i,j} - \sum_j \mathbf{q}_{i,j}\right\| \\ &\le n\|\widehat{\mathbf{q}}_{i,j} - \mathbf{q}_{i,j}\|.\end{aligned}$$

$$(57)$$

$\square$

## E.2 PROOF OF LEMMA 1

Here we prove Lemma 1.

*Proof.* Equation (9) says

$$\mathbf{q}_{t,i} \coloneqq g\left(\{\mathbf{q}_{t,i,j}\}_{j=1}^n\right) = \sum_j I_{i,j}\mathbf{q}_{t,i,j},$$

where $I_{i,j}$ is an indicator function to denote if two agents are connected. Under the assumption that the static non-complete graph has maximum of number of degrees $k$ and the agents are homogeneous, we have $\sum_j I_{i,j} <= k$. Thus,

$$\begin{aligned} \|\widehat{\mathbf{q}}_i - \mathbf{q}_i\| \quad &= \left\|\sum_j I_{i,j}\widehat{\mathbf{q}}_{i,j} - \sum_j I_{i,j}\mathbf{q}_{i,j}\right\| \\ &\leq k\left\|\widehat{\mathbf{q}}_{i,j} - \mathbf{q}_{i,j}\right\|. \end{aligned} \tag{58}$$

$\square$

### E.3 INFORMAL PROOF OF LEMMA 2

Here we make some intuitions for the proof of Lemma 2. As we already give the proof sketch in our main paper. Lemma 2 is a direct application of Theorem 2. Here our random variable $J_1, \cdots, J_n \sim \mathcal{NB}(r, p)$, where $\{(J_i)\}_{i=1}^n$ represents the number of time points node $i$ needs before it meets node $j$ number of $r$ times.

For the complete static graph, we need at least $N$ sample for the bound in equation 8 to be valid; in other words, we need the trajectory to run at least $N$ time points to collect enough data to estimate our conditional operator accurately $\mathbf{q}_{i,j}$.

For the dynamic graph, each time $t$, the two nodes are randomly connected with probability $p$, if it connects, then we can obtain a valid sample to estimate $\mathbf{q}_{i,j}$; if not, then we skip to the next time step. For the dynamic graph node $i$, if we take the union set of the nodes $i$ connected over the trajectory path, then the union set could form a static complete graph. We say the two graphs are equivalent. The number of time points needed by $i$ until the $N^{th}$ connection with $j$ for each of the pairs $(i, j)$ follows the same distribution $J \sim \mathcal{NB}(N, p)$. Then we are interested in the expectation of the maximum of $J_1, \ldots, J_{n-1}$, we denote it as $J_{\{1,\ldots,n-1\}}$.

$\mathbb{E}\{J_{\{1,\ldots,n-1\}}\}$ means on average, how many time points $(N')$ we need for all nodes other than $i$ at least meets $N$ times with $i$. Obviously, this $N' >= N$ since $p \in [0, 1]$. The calculation of this expectation is solved by (Grabner & Prodinger, 1997). And we also put their result in Theorem 2.

