# OpenReview forum: "Reinforcement Learning under a Multi-agent Predictive State Representation Model: Method and Theory"
_ICLR.cc/2022/Conference — ICLR 2022 Spotlight_

### Official Review · Reviewer_6qEL · 2021-11-02

**Correctness:** 3
**Technical Novelty And Significance:** 4
**Empirical Novelty And Significance:** 3
**Recommendation:** 8
**Confidence:** 3

**Main Review:**

Strengths:
The paper proses a novel and complete PSR-based MARL method which also scales well with the use of graph structures. The performance of the MAPSR is analized both theoretically and empirically.  The quality of the learned policies are also verified by experiments.

Weaknesses:
I think the paper overall is good. It would be better to see some extensions or future thoughts on the unknown interaction structures which are not always trivial and need to be learned.

**Summary Of The Paper:**

The paper introduces a predictive state representation (PSR)-based MARL framework. The framework uses a graph representation to model the interactions between agents. Performance bounds are given for the learned predictive state representation.  With the MAPSR, individual policies are trained by replacing the partial observation o with PSR Q. The experiments results show the advantage of the proposed method compared with the baselines experimented in the paper.

**Summary Of The Review:**

The paper gives solid technical contributions and substantial experiments. I recommend acceptance of the paper.

---

> ### Author Response · Authors · 2021-11-18
> **Response to Reviewer 6qEL**
>
> We thank Reviewer 6qEL for the detailed and helpful feedback. We are encouraged that Reviewer 6qEL thinks our paper is of solid technical merits and our experimental results are substantial.
>
> We have made the claims more clear as suggested by Reviewer 6qEL. Below we address Reviewer 6qEL's questions in detail.
>
> > Q1: I think the paper overall is good. It would be better to see some extensions or future thoughts on the unknown interaction structures, which are not always trivial and need to be learned.
>
> - Thank you for this suggestion. Although out of the scope of this paper, we agree that extending to an unknown interaction structure is an important open question.
>
> - When the interaction structure is unknown, we can still seek to model the relationship between agents via a graph. But the difficulty is that we need to learn this graph.
>     -  The generative model with the recursive structure may work. We could use a neural network that is trained via back-propagation to output a graph $G$ (the binary matrix $I_{t}$ in our paper).
>     -  Here we give an approachable example: In the coordinate-grid world, assume we have a spatial state, which could be a coordinate grid with two perpendicular axes ($x$-axis, $y$-axis). Each agent can be identified by an ordered pair of numbers ($x$-coordinate, $y$-coordinate): $\{(x,y) \in \mathcal{X} \times \mathcal{Y}, \mathcal{X} \subset N,  \mathcal{Y} \subset N\}$, that indicates where agent's location in the coordinate grid. We can represent the spatial state as a tensor $T \in \mathbb{R}^{|\mathcal{X}| \times |\mathcal{Y}| \times |K|}$, where the first two dimensions represent grids $x$, $y$ coordinates, and last one dimension represents the agent index. $K$ is the set that contains the agent index, the total number of agents is $n$ ($|K| = n$). So $T_{i,j,k} = 1$ if and only if agent $k$ is present in position $(i,j)$ of the coordinate grid. As time goes, we have a sequence of $T_1, T_2, \cdots, T_t$ that are collected at different time points. We can let the inputs of neural network be the current spatial state $T_t$ and some previous time spatial states $\{T_{t-5}, \dots, T_{t-1}\}$, agent actions, and last time graph $G_{t-1}$. Then the forward pass is carried via a convolution, a pooling, padding, and some fully connected layers, and then a graph generation network. The graph generation network can be a self-attention [1] network with a recurrent structure that uses an attention mechanism to catch long and short-term time dependence. The result of it will be the graph $G_t$ in the current time. The loss can be the squared norm difference between two matrices. We can set this graph generating training ahead, then add this pre-trained network into our framework.
>
> - The idea of generating a graph representing the relationship between agents when the graph is unknown by training a neural network with an attention mechanism and incorporation into the decision process is also presented in the work of [2].
>
> [1]Vaswani, Ashish, et al. "Attention is all you need." Advances in neural information processing systems. 2017.
>
> [2]Malysheva, Aleksandra, et al. "Deep multi-agent reinforcement learning with relevance graphs." arXiv preprint arXiv:1811.12557 (2018).
>
> > Q2: Some of the paper’s claims have minor issues. A few statements are not well-supported or require small changes to be made correctly.
>
> We thank the reviewer for making this suggestion. We have double-checked the manuscript carefully and addressed the problems as much as possible by providing more references and making clearer statements.
>
> We have made some modifications to further improve the accessibility of the paper: we moved POMDP in the first paragraph to the related work, we added relevant explanations to some notations, we added background review of PSR and multi-agent RL in the appendix, and we proofread the manuscript multiple times during this discussion period.
>
> We will continue polishing the writing and fixing grammatical errors.
>
>
> ---
> We greatly appreciate Reviewer 6qEL's positive comments and constructive suggestions. We have updated our paper to address all the points raised by the reviewer. Please let us know if there are other questions or concerns. Thank you!

---

### Official Review · Reviewer_Jba3 · 2021-11-02

**Correctness:** 4
**Technical Novelty And Significance:** 4
**Empirical Novelty And Significance:** Not applicable
**Recommendation:** 8
**Confidence:** 3

**Main Review:**

The work is clearly explained an the mathematical results and definitions are appropriately used and of a very high quality (by my assessment. One criticism I have is that some terminology could be introduced at the outset to assist the reader (see detailed comments). Another issue is that the language and phrasing is sometimes a little odd and the paper would benefit from a quick grammar check.

Nonetheless, the motivation is very clear to me. The idea of efficiently learning PSRs is a powerful one, and this work applies it to the multi-agent domain introducing some very general ways to account for the other agents' influences on the observations of a focal agent. One repeated theme in recent scalable MARL approaches is how to manage the apparent non-stationarity from the perspective of one agent induced by the other agents adapting alongside. This approach is arguably a natural way to do this, in terms of action conditioned predictions of observations and how these are influenced by other agents.

The methods are evaluated against some very sensible benchmarks, including a PSR method that omits the information sharing between agents part, and a direct independent actor critic that doesn't incorporate PSRs at all. As such the empirical evaluation seems to be sensible and robust to me. Results show a significant improvement over the methods without the newly proposed PSR learning components.

I am satisfied that the work is of a very high quality as far as I can assess it.

## More detailed comments

It's not clear what is meant by this:

> Learning in this environment is challenging since
these partially observable observations bring the noise.

Or this:

> in the setting where the system model is not known a priori or the
model introduced by the domain expert is biased, it is always beneficial and safer to learn system models and develop policies based on the models.

Is it always beneficial to learn POMDP models, or can model free methods sometimes work more effectively?


In section 3 we start working with operators, and the notation is necessarily quite dense in order to get all the definitions and results in but it would assist the reader to have some guidance on how to read things on first presentation. For instance at the bottom of p3 where you say:
> In this model, predictive state Q t satisfies...

It would be helpful to know that $Q_t$ and $P_t$ are conditional expectation operators. The $\circtimes$ operator at the top of p4 could be better introduced too. I am guessing this is the kronecker product.


The phrasing is sometimes a little odd and the paper would benefit from a quick grammar check. For instance, there are a number of instances of things like this:
> We replace the partially observation o with PSR Q

I am not sure I fully follow what is being shown in Figure 2. Perhaps and example of how to decode one of these plots would help.

**Summary Of The Paper:**

The authors present a framework and method in which predictive state representations for multiple agents simultatneously acting and interacting within an environment. This is presented in a general way, where predictive states are Hilbert space operators which when applied to sequences of observations and actions appropriately predict the predictions of these state representations. The key advance in this work is to apply this existing PSR framework to networks of agents, with 3 types of agent network: static complete graph (all agents affect all others experience); static non-complete graph (only some agents affect one another); and dynamic non-complete graph (agents affect one another in a time varying way). A number of theoretical results are presented, including PAC bounds for the approximators in the framework.

The authors then present two closely related methods to learn policies alongside these multi agent PSRs in an online way. The first MAPSRL-1 is akin to the independent actor critic (Foerster et al., 2017) and thus may suffer from the apparent non-stationarity of the environment from the perspective of any one agent. MAPSRL-2 addresses this by incorporating the PSR information from other agents into the policy gradient update.

The paper presents a series of experiments based on environments encoded in the  OpenAI Gym MAMujoco system. These are environments presented in previous papers and there are a broad selection of these.


**Summary Of The Review:**

The paper presents a clearly motivated and well founded approach for making multi-agent RL more stable and robust (in particular to non-stationarity) by use of predictive state representations that incorporate inter-agent interactions. In my opinion this paper is of very high quality, relevance and novelty.

---

> ### Author Response · Authors · 2021-11-18
> **To Reviewer Jba3: Clarification of a claim about noisy observation.**
>
> We thank Reviewer Jba3 for the detailed and insightful feedback. We are happy that Reviewer Jba3 finds our paper well-motivated, our analysis sound, and our experiments extensive.
>
> We have updated our manuscript and made the claims clearer, as suggested by Reviewer Jba3. Below we address Reviewer Jba3's concerns in detail.
>
> > Q1: It's not clear what is meant by this: "Learning in this environment is challenging since these partially observable observations bring the noise."
>
>
> Here we mean: Learning in this environment is challenging since partial observations are contaminated by the noise in the observation channel. In POMDP, an agent does not observe the state itself, instead it gets sensory measurements from a so-called observation channel $\Omega:\mathcal{S} \times \mathcal{A} \mapsto \Pi({\mathcal{O}})$ ($\mathcal{S}$ is the finite set of states of the world, $\mathcal{A}$ is a finite set of actions, $\mathcal{O}$ is a finite set of observations the agent can observe), which defines a function with $\Omega(o, s', a) = P(o|a,s')$, the conditional probability of observing $o$ after taking action $a$ and transition to state $s'$.
>
> $\Omega$ characterizes the sensor noise, and different observations can be perceived even in the same state. Due to state aliasing and sensor noise, an observation only provides partial information about the world state, hence it cannot be directly used as the information state for planning. Instead, the observation provides some evidence of the current state and allows the uncertainty of the world state to be updated. The uncertainty of the world state is characterized by the belief state, which is the sufficient statistics of observation history[1].
>
> Thank you very much for raising this question, we have updated the manuscript. We defer the discussion of POMDP to the related work section.
>
> [1] R. D. Smallwood and E. J. Sondik. The optimal control of partially observable Markov processes over a finite horizon. Operational  Research,  21:1071–1088, 1973.

---

> > ### Author Response · Authors · 2021-11-18
> > **To Reviewer Jba3: Clarification of a claim about model-based vs model-free learning.**
> >
> > > Q2: It's not clear "In the setting where the system model is not known a priori or the model introduced by the domain expert is biased, it is always beneficial and safer to learn system models and develop policies based on the models." Is it always beneficial to learn POMDP models, or can model-free methods sometimes work more effectively?
> >
> >
> > - In POMDP, the agents are unsure which state they are in. Model-free RL methods directly rely on the experience of agent-environment interactions and use the observations to learn the policy.  Early work for model-free RL includes U-tree [1] and Monte Carlo based reactive policy (MCESP) [2].  Recent model-free deep RL such as [6,7,8] unifies representation and task learning into a single end-to-end training procedure, but solving both problems together is difficult since they are interleaved, so directly learning from standard model-free RL algorithms in the high-dimensional input space could in proactive be slow, sensitive to hyperparameters and inefficient.
> >
> >
> > - On the other hand, modeling the system dynamics could enhance sample efficiency by searching the policy under a fitted model that approximates the true dynamics. Therefore, **model-based approach** is favored in problems where a limited number of iterations are available like arduous processes and continuous robotic control.
> >
> >
> > - **Here we meant to say: especially in the circumstances when learning the model is beneficial (i.e., a limited number of interactions with the environment available, high-dimensional observations, continuous robotic control) and the model is not known, it is beneficial to learn the model for the system with a compact representation. The model should represent the system dynamics well but also not be too complicated to learn.**  Reviewer is right that it is not always beneficial to learn POMOP models; actually, solving POMDP optimally is hugely intractable, and if the above constraints mentioned are relieved, the model-free methods will work more efficiently. We appreciate the reviewer’s question and we have updated the manuscript accordingly.
> >
> >
> > - **Besides, since our paper does not focus on POMOP, we moved the POMDP part to the related work section.**
> >
> > [1] A. K. McCallum. Reinforcement learning with selective perception and hidden state.  PhD thesis, University of Rochester, 1996.
> >
> > [2] T. J. Perkins. Reinforcement learning for POMDPs based on action values and stochastic optimization.  In AAAI/IAAI, pages 199–204, 2002.
> >
> > [3] N.  Meuleau,  L.  Peshkin,  K.-E.  Kim,  and  L.  P.  Kaelbling.   Learning finite-state controllers for partially observable environments. In UAI , pages 427–436. Morgan Kaufmann Publishers Inc., 1999.
> >
> > [4] D. Silver and J. Veness.  Monte-Carlo planning in large POMDPs.  In NIPS, volume 23, pages 2164–2172, 2010
> >
> > [5] A. Somani, N. Ye, D. Hsu, and W. S. Lee. DESPOT: Online POMDP planning with regularization.  In NIPS, pages 1772–1780, 2013.
> >
> > [6] M. Hausknecht and P. Stone. Deep recurrent Q-learning for partially observable MDPs. In AAAI Fall Symposium on Sequential Decision Making for Intelligent Agents, 2015.
> >
> > [7] J. Foerster, I. A. Assael, N. de Freitas, and S. Whiteson. Learning to communicate with deep multi-agent reinforcement learning. In Neural Information Processing Systems (NIPS), 2016.
> >
> > [8] P. Zhu, X. Li, P. Poupart, and G. Miao. On improving deep reinforcement learning for POMDPs. arXiv preprint arXiv:1804.06309, 2018.

---

> > > ### Author Response · Authors · 2021-11-18
> > > **To Reviewer Jba3: response to some wording related suggestions and explanation of Fig 2.**
> > >
> > > > Q3: "In section 3 we start working with operators, and the notation is necessarily quite dense in order to get all the definitions and results in but it would assist the reader to have some guidance on how to read things on the first presentation. For instance at the bottom of p3 where you say:
> > > In this model, predictive state Q t satisfies… It would be helpful to know that  Qt and  Pt are conditional expectation operators. The \circtimes operator at the top of p4 could be better introduced too. I am guessing this is the Kronecker product."
> > >
> > > Thank reviewer Jba3 for the advice. We have added more explanations to them.
> > >
> > > 1.  In this model, predictive state $Q_{t}$ satisfies $Q_{t} \psi_{t}^a = \mathbb{E}[\psi_{t}^o|\psi_{t}^a; \psi_{t}^h]$  and extended predictive state $P_{t}$ satisfies $P_{t}\xi_{t}^a = \mathbb{E}[\xi_{t}^o|\xi_{t}^a;\psi_{t}^h]$ (i.e., $Q_{t}$ and $P_{t}$ are conditional linear expectation operators which map to the conditional expectation of future observations).
> > >
> > > 2.  We use $\otimes$ to denote the transposed Khatri–Rao product for two matrices with the same number of rows, and each row of the resultant matrix is the vectorized outer product of the corresponding row vectors in the two matrices.
> > >
> > > We will make the description more clear in the revised paper.
> > >
> > >
> > > > Q4: The phrasing is sometimes a little odd and the paper would benefit from a quick grammar check. For instance, there are a number of instances of things like this:
> > > "We replace the partially observation o with PSR Q"
> > >
> > >
> > > More precisely, we mean: Here, the predictive states $Q_i$ estimated by the MAPSR are considered as states to fit the value and policy functions.
> > >
> > > Thank you for your suggestion. We have made some modifications to further improve the accessibility of the paper: we moved POMDP in the first paragraph to the related work, we added relevant explanations to some notations, we added background review of PSR and multi-agent RL in the appendix, and we proofread the manuscript multiple times during this discussion period.
> > >
> > > We will continue polishing the writing and fixing grammatical errors.
> > >
> > >
> > > > Q5: I am not sure I fully follow what is being shown in Figure 2. Perhaps an example of how to decode one of these plots would help.
> > >
> > >
> > > >
> > > Thank you for your question. Figure 2 shows the predicted one-step observations calculated by $\hat{o}_{t} = Z(Q_t, a_t)$ for every agent $i$, vs. the observations the agent has actually seen collected from one trajectory (many steps) during a training iteration. Let’s use figure 2(f) of 4-agents Ant as an example. Each column corresponds to an agent and each row represents a coordinate of the observation vector. Since we work on the partially observable environment, we set each agent to see three coordinates among the full observation list containing their 3D positions, joint angles, etc. The x-axis denotes the trajectory length, and the y-axis represents the iteration number. The black dot is the actual observation emitted by the environment after receiving an action, while the colored line is the predicted observation.
> > >
> > >
> > > ---
> > > We greatly appreciate Reviewer Jba3 for the positive comments and constructive suggestions. We have updated our paper to address all the points raised by the reviewer. Please let us know if there are other questions or concerns. Thank you!

---

> > > > ### Comment · Reviewer_Jba3 · 2021-11-29
> > > > **Thank you**
> > > >
> > > > As I stated in the original review, I believe the paper to be of high quality. The changes made by the authors in light of the reviewer comments seem sensible and strengthen the paper further. I continue to champion this paper for acceptance.

---

### Official Review · Reviewer_zgHd · 2021-11-04

**Correctness:** 4
**Technical Novelty And Significance:** 3
**Empirical Novelty And Significance:** 4
**Recommendation:** 8
**Confidence:** 3

**Main Review:**

Strength:
The mathematical derivation appears sound and theoretical guarantee is given.
The two versions of the algorithms, MAPSR1 and 2, performed well in MAMuJoCo benchmark.
Weakness:
It needs a lot of background knowledge about PSR, MARL and graph-based approaches to understand the work, though that is inevitable...
Regarding statements like "PSR is more compact than POMDP": POMDP is a problem setting and PSR is one way to solve that. It makes better sense if you specify "belief-state approaches to POMDP".
The reference source for Littman et al. 2001 (NIPS 2001?) is missing.


**Summary Of The Paper:**

This paper extends predictive state representation (PSR) for POMDP to multi-agent reinforcement learning (MARL) setting.

**Summary Of The Review:**

This paper presents practical algorithms for MARL by utilizing the data-driven dynamic modeling by PSR.

---

> ### Author Response · Authors · 2021-11-18
> **Response to Reviewer zgHd**
>
> We thank Reviewer zgHd for the detailed and insightful feedback. We are encouraged that Reviewer zgHd thinks our paper is mathematically and theoretically sound. We have updated our manuscript and made the claims more clear as suggested by Reviewer zgHd. Below we address Reviewer zgHd's suggestions in detail.
>
> > Q1: It needs a lot of background knowledge about PSR, MARL, and graph-based approaches to understand the work, though that is inevitable.
>
> Thank you for your suggestions, they are really helpful. We have followed your suggestions and provided a more background knowledge review in the appendix of the revised paper.
> - In appendix A.1, we add the mathematical model of PSR, which we believe can be helpful for people who are not readily familiar with the PSR model.
> - In appendix B.1.1, we add the multi-agent MDP model, and in Appendix B.1.2, we add the multi-agent POMDP model. We summarize the two MARL algorithms IAC and MADDPG, which are relevant to our method, in appendix B.2.1 and B.2.2, respectively.
> - In section 4, we show how we model the multi-agent by a graph network. Also, in the first paragraphs of sections 4.1, 4.2, and 4.3, we introduce the structures of different types of graphs.
>
> > Q2: Regarding statements like "PSR is more compact than POMDP": POMDP is a problem setting, and PSR is one way to solve that.
>
> Here we mean that PSR offers a more compact state representation for POMDP models.  In particular, Theorem 1 in [1] shows that for any finite POMDP, there exists a linear PSR with strictly more compact representation. We thank Reviewer zgHd for this question. We have rewritten that sentence to make it more precise.
>
> [1] Littman, Michael L., Richard S. Sutton, and Satinder P. Singh. "Predictive representations of state." NIPS. Vol. 14. No. 1555. 2001.
>
> > Q3: It makes better sense if you specify "belief-state approaches to POMDP".
> Yes, we refer to the common approaches of solving the POMDP by lifting the original POMDP to an MDP whose states are belief states. In other words, the POMDP is cast into an MDP where belief states comprise the MDP state space, which is called "belief MDP". Therefore, the belief MDP is a quadruple $<\mathcal{B}, \mathcal{A}, T^b, R^b>$.
> - $\mathcal{B} = \Pi(\mathcal{S})$ is the set of belief states.
>    Here a belief state $b(s)\in \mathcal{B} = Pr(s|o, a, b)$ is a posterior distribution over states given an observation $o$, an action $a$, and an old belief state $b$. So the belief state space forms a distribution space over possible states.
>    - Computing the belief states requires a Bayesian update.
> $$b(s’) = Pr(s’|o,a,b) = \frac{Pr(o|s’,a,b)Pr(s’|a,b)}{Pr(o|a,b)}\\
> = \frac{Pr(s’,a,o) \sum_{s\in \mathcal{S}} Pr(s’|a,b,s) b(s)}{Pr(o|a,b)} \\
> = \frac{\Omega(s’,a,o) \sum_{s\in \mathcal{S}} T(s,a,s’) b(s)}{Pr(o|a,b)}$$
>    - Here $\Omega: \mathcal{S} \times \mathcal{A} \mapsto \Pi(\mathcal{O})$ is an observation function (observation channel) that calculates the probability that the observation $o$ will be recorded after an agent performs action $a$ and lands in state $s’$.
>    - $T:\mathcal{S} \times \mathcal{A} \mapsto \Pi(\mathcal{S})$ is the state-transition function, that computes the probability of ending in state $s’$, given that agent starts in state $s$ and takes an action $a$.
> - $\mathcal{A}$: the set of actions for the agent, remains the same in the POMDP.
> - $T^b: \mathcal{B}\times \mathcal{A} \mapsto \mathcal{B}$ is the belief transition function:
> $$T^b(b,a,b’) = Pr(b’|b,a) = \sum_{o\in\mathcal{O}} Pr(b’|a,b,o) Pr(o|a,b)\\
> = \sum_{o\in\mathcal{O}} Pr(b’|a,b,o) \sum_{s’\in \mathcal{S}}\Omega(s’,a,o)\sum_{s\in\mathcal{S}}T(s,a,s’)b(s)$$
> Where $Pr(b’|a,b,o) = 1$ if and only if the updated belief state based on the last action, the current observation, and the previous belief state has its value equals to $b’$.
> - $R^b: \mathcal{B}\times \mathcal{A} \mapsto \mathbb{R}$
> The reward function, $R^b(b,a) = \sum_{s\in\mathcal{S}} b(s) R(s,a)$. The reward function represents the true expected reward to the agent given that belief state represents the true probabilities for all states $s\in \mathcal{S}$.
>
> In the belief MDP, the agent uses a policy $\pi:\mathcal{B} \mapsto \mathcal{A}$ that maps belief states to actions to receive actions. The agent executes actions prescribed by the policy, then updates its belief state (probability distribution over the system states). The goal for the agent is to find an optimal policy that gives the maximum expected sum of rewards over the horizon.
>
> PSR is a method that learns a compact representation of belief states.
>
> We thank the Reviewer for making this suggestion. In addition, we have updated the literature review on belief-state approaches to POMDP in the related work section of our manuscript.

---

> > ### Author Response · Authors · 2021-11-18
> > **Response to Reviewer zgHd**
> >
> > > Q4: The reference source for Littman et al. 2001 (NIPS 2001?) is missing.
> > >
> > This is indeed a reference compilation issue. Thank Reviewer zgHd for pointing it out. We have fixed it.
> >
> > ---
> > We greatly appreciate Reviewer zgHd for the positive comments and constructive suggestions. We have updated our paper to address all the points raised by the Reviewer. Please let us know if there are other questions or concerns. Thank you!

---

### Author Response · Authors · 2021-11-22
**General Response: Summary of Paper Updates**

We thank all reviewers for their recognition of our work. All their comments are helpful and have been reflected in the updated paper.

Here we briefly outline the updates to the revised submission based on the reviews. Then, we address individual questions of reviewers in different responses.

### Paper Updates

- **[Section 1]**: (1) We described the PSR and POMDP more precisely. (zgHd) (2) We deferred the discussion of POMDP to the related work section, and we made the description of claims clearer. (Jba3)
- **[Section 2]**: We added more descriptions about the belief state. We cited more papers on "belief state approaches to POMDP. (zgHd)
- **[Section 3]**: We have expanded the notations of PSR model in greater detail, including the meaning of predictive states $Q_t$ and extended predictive states $P_{t}$, and the meaning of $\otimes$. (Jba3)
- **[Section 5]**: We made the description of the value function more precise. (Jba3)
- **[References]**: We updated the publication venues of some references. (zgHd)
- **[Appendix]**:  In A.1, we have added the background of the mathematical model of PSR. In B.1,  we have added the description for the Multi-agent MDP model, the multi-agent POMDP model, respectively. In B.2: We summarized the two MARL algorithms IAC and MADDPG. (zgHd)
- **[The entire paper]**: We improved the writing and made the description of claims more precise. (6qEL) We fixed several typos, reference compiling issues. We proofread the paper to make sure the paper was without any grammar errors. (Jba3)

We greatly appreciate all reviewers' suggestions. Furthermore, we hope that our paper updates and responses have addressed reviewers' questions and concerns. Please let us know if there are further questions.

---

### Decision · Program_Chairs · 2022-01-20

**Decision:**

Accept (Spotlight)

**Comment:**

This paper presents an extension of the Predictive State Representation (PSRs) to multi-agent systems, with a dynamic interaction graph represents each agent’s predictive state based on its “neighborhood” agents. Three types of agent networks are considered: static complete graphs (all agents affect all others experience); static non-complete graphs (only some agents affect one another); and dynamic non-complete graphs (agents affect one another in a time varying way). A number of theoretical results are presented, including PAC bounds for the approximations in the framework. The paper also contains a number of experiments that clearly show the advantages of the proposed technique over some related methods.

The reviewers unanimously agree that this is a strong paper, with a solid theoretical and empirical analysis.